# 4M: Massively Multimodal Masked Modeling

**David Mizrahi**[1,2*]    **Roman Bachmann**[1*]    **Oğuzhan Fatih Kar**[1]
**Teresa Yeo**[1]   **Mingfei Gao**[2]   **Afshin Dehghan**[2]   **Amir Zamir**[1]
[1]Swiss Federal Institute of Technology Lausanne (EPFL)    [2]Apple

https://4m.epfl.ch

## Abstract

Current machine learning models for vision are often highly specialized and limited to a single modality and task. In contrast, recent large language models exhibit a wide range of capabilities, hinting at a possibility for similarly versatile models in computer vision. In this paper, we take a step in this direction and propose a multimodal training scheme called 4M. It consists of training a **single unified Transformer encoder-decoder** using a **masked modeling objective** across a **wide range of input/output modalities** – including text, images, geometric, and semantic modalities, as well as neural network feature maps. 4M achieves **scalability** by unifying the representation space of all modalities through mapping them into discrete tokens and performing multimodal masked modeling on a small randomized subset of tokens.

4M leads to models that exhibit several key capabilities: (1) they can perform a diverse set of vision tasks out of the box, (2) they excel when fine-tuned for unseen downstream tasks or new input modalities, and (3) they can function as a generative model that can be conditioned on arbitrary modalities, enabling a wide variety of expressive multimodal editing capabilities with remarkable flexibility.

Through experimental analyses, we demonstrate the potential of 4M for training versatile and scalable foundation models for vision tasks, setting the stage for further exploration in multimodal learning for vision and other domains.

## 1   Introduction

In recent years, the field of natural language processing (NLP) has seen a shift toward training large language models (LLMs) that are inherently capable of performing a wide range of tasks without requiring extensive task-specific adaptations [12, 25]. While these models have demonstrated remarkable success in NLP, there remains a need to develop similarly versatile and scalable models for vision. A crucial aspect of scalability and versatility in vision is the ability to handle multiple (input) modalities and (output) tasks, as vision models must deal with a diverse range of sensory inputs, such as images, 3D, and text and solve a wide range of tasks.[†] Unlike NLP, where language modeling on raw text has led to multitask capabilities [84, 86], training on only RGB images with a single objective has not exhibited the same behavior for vision. Therefore, it is deemed important to incorporate multiple modalities and tasks in training. It has been indeed suggested by psychophysical studies that multimodality is one key driver behind the development of biological intelligence [104].

To create a model that exhibits the desirable properties of foundation models in vision, it is important to consider three key aspects in terms of scalability: data, architecture, and training objective. For data, scalability means being able to benefit from more training samples toward improving performance.

---

[†]For clarity, "modalities" usually denote the *inputs* to a model (e.g. sensory signals), and "tasks" usually denote the *outputs* (e.g. semantics). Our method enables a symmetric input-output structure, thus we use "modalities" and "tasks" interchangeably in this paper.

[*]Equal contribution & corresponding authors.

37th Conference on Neural Information Processing Systems (NeurIPS 2023).

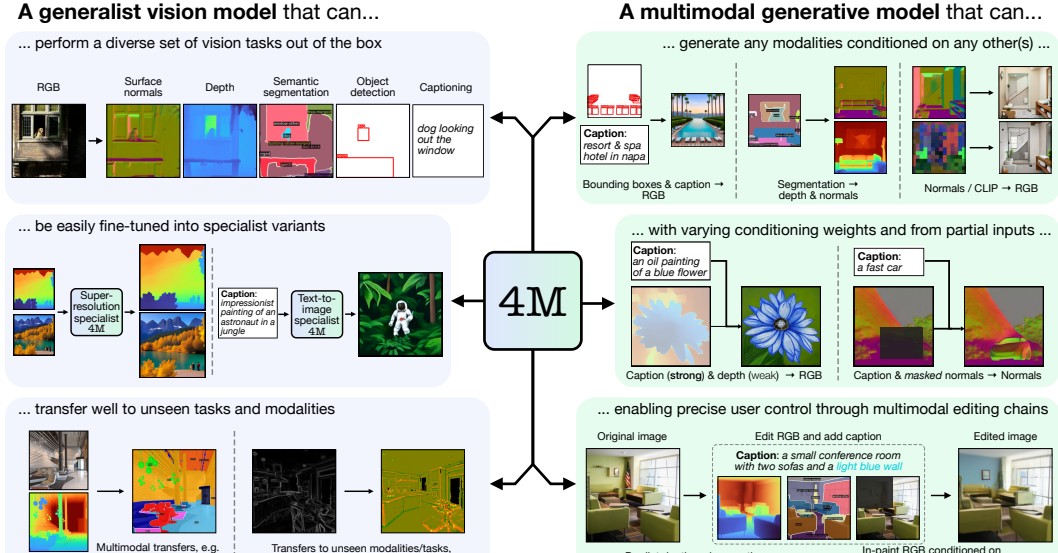

Figure 1: 4M enables training a versatile multimodal and multitask model, capable of **performing a diverse set of vision** tasks out of the box, as well as being able to perform **multimodal conditional generation**. This, coupled with the model's ability to perform in-painting, enables powerful image editing capabilities. This generalist model transfers well to a broad range of downstream tasks or to novel modalities, and can be easily fine-tuned into more specialized variants of itself.

In terms of architecture, scalability implies increased performance with growing model size and remaining stable when trainingd at large sizes. Lastly, a scalable training objective should efficiently handle a growing number of modalities without incurring excessive computational costs. In our approach, we target scalability across these three aspects while *maintaining compatibility* with multiple modalities.

We address these challenges by proposing a method consisting of training a single unified Transformer encoder-decoder using a multimodal masked modeling objective. We name this approach 4M (short for "**M**assively **M**ultimodal **M**asked **M**odeling")[‡] to emphasize its ability to scale to *many diverse modalities*. Our method unifies the benefits of multimodal learning and masked modeling, such as (1) serving as an effective pre-training objective for learning rich representations [30, 48], (2) leading to strong cross-modal predictive coding abilities and shared scene representations [5, 44], (3) enabling models to be used for generative tasks through iterative sampling [17, 18]. Crucially, 4M combines these benefits while remaining *efficient* via a number of mechanisms.

To enable training a single Transformer on modalities with different formats, like text, bounding boxes, images, or neural network features, we choose to unify their representational spaces by mapping them into sets or sequences of discrete tokens [21, 22, 74, 64] using modality-specific tokenizers [110]. This tokenization approach enhances compatibility, scalability, and sharing by removing the need for task-specific encoders and heads, allowing the Transformer to be compatible with all modalities and maintain full parameter-sharing. Also, although 4M operates on a large set of modalities, it can train in a highly efficient manner through the use of *input* and *target masking*. This involves randomly selecting a small subset of tokens from all modalities as *inputs* to the model, while another small subset of the remaining tokens is treated as *targets*. Decoupling the number of input and target tokens from the number of modalities prevents the computational cost from rapidly escalating with increasing modalities, allowing for a scalable training objective.

Leveraging the availability of single-modal or text-image pair datasets, such as CC12M [19], we employ strong pseudo labeling networks to generate aligned binding data across various modalities. This pseudo labeling approach enables training on diverse and large-scale datasets without demanding them to come with multimodal/multitask annotations.

4M models are capable of performing many key vision tasks out of the box and can also be fine-tuned to achieve highly competitive performance on unseen downstream tasks and input modalities. In

---

[‡]We sometimes also refer to the *trained models* as 4M (short for "**M**assively **M**ultimodal **M**asked **M**odel").

addition, training using a multimodal masked modeling objective leads to *steerable* generative models that can be conditioned on arbitrary modalities, enabling the user's intent to be expressed in a versatile manner (as depicted in Figure 4) as well as various multimodal editing tasks (see Figure 1).

We further perform an extensive ablation analysis that studies the factors affecting 4M's performance. This thorough examination, along with the simplicity and generality of our approach, demonstrates the potential of 4M for a wide range of vision tasks and further expansions.

Our main contributions and results can be summarized as follows:

1. **Method**: we introduce 4M, a framework for training versatile and scalable foundation models for vision tasks using a multimodal masked modeling objective. Our approach results in models that learn rich representations and perform well on a wide range of tasks without requiring task-specific adaptations.
2. **Performance**: we demonstrate the efficacy of our approach through extensive experiments and benchmarks, showcasing the ability of these models to perform many key vision tasks out of the box, as well as achieving highly competitive performance when fine-tuned on unseen downstream tasks.
3. **Generative Capabilities**: we showcase the flexible and steerable generative capabilities of models trained using 4M, enabling a variety of multimodal editing tasks utilizing conditioning on arbitrary modalities.
4. **Experimental Study**: we conduct an extensive ablation analysis to study the factors affecting 4M's performance, providing important insights into these models' behavior and design.

Code, models, and additional interactive visualizations are available at `https://4m.epfl.ch`.

## 2 Method Description

The 4M architecture and training objective (depicted in Figure 2) were designed with a focus on being as *compatible* and *scalable* as possible in terms of the number and type of modalities it accepts, while being conceptually *simple* and computationally *efficient*. We enable these through the conjunction of the following key aspects:

1. **Tokenizing modalities**: We abstract away modality-specific intricacies by mapping all modalities into sequences or sets of discrete tokens, whether they are images, text, sparse data, or neural network feature maps. This allows every possible mapping between modalities to be seen as predicting one sequence or set of tokens from another. In Section 2.1, we discuss what types of modalities we train on, how we generate the training data, and how we enable training a model on different modalities through tokenization.
2. **Training a single compatible network on all modalities**: Different tasks in vision, NLP and other domains traditionally required vastly different modeling choices, architectures, and losses, making the joint training on multiple modalities challenging. Tokenizing all modalities into a unified representation space allows us to train a single Transformer encoder-decoder (see Figure 2) to map between different modalities through (parallel or serialized autoregressive) token prediction. In Section 2.2, we provide more details on the 4M architecture.
3. **Multimodal masked pre-training objective**: Transformers have demonstrated excellent scalability with data and model size across a diverse set of tasks [60, 2], particularly when paired with a scalable pre-training objective such as masked reconstruction [30, 48, 56]. In Section 2.3, we detail our approach to training 4M using a multimodal masked modeling objective on randomized token subsets to learn strong cross-modal predictive coding abilities.

### 2.1 Modalities & data

**Pre-training modalities.** We train 4M models on a diverse set of modalities, namely RGB, captions, depth, surface normals, semantic segmentation maps, bounding boxes, and tokenized CLIP feature maps [85, 82, 114]. These modalities were chosen to cover several key aspects: First, they contain a mix of semantic information (captions, semantic segmentation, bounding boxes, CLIP), geometric information (depth, surface normals), and RGB. When used as input modalities, these modalities can be used as informative priors about the scene geometry and its semantic content [98, 69], and when used as target tasks, they allow us to steer what kind of representations are learned [42, 5, 82, 114]. Second, these modalities are diverse in terms of the format they use to encode information. They consist of dense visual modalities (RGB, depth, surface normals, semantic segmentation), sparse and/or sequence-base modalities (captions, bounding boxes), as well as neural network feature maps (CLIP [85]). Finally, these modalities allow for diverse and rich interaction with the model for

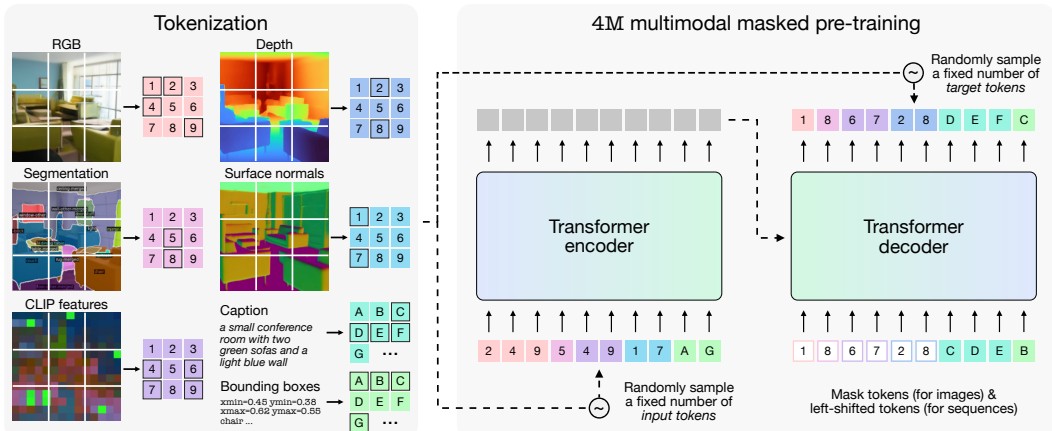

Figure 2: **Method overview.** (Left): 4M is a framework for training multimodal and multitask models that operate on tokenized versions of multiple image-like modalities (such as RGB, depth, etc.) and sequence modalities (such as captions and bounding boxes). (Right): The 4M pre-training objective consists of training a Transformer encoder-decoder to predict a randomly selected subset of tokens, which is sampled from all modalities, based on another random subset of tokens.

generative purposes. For example, captions, segmentation maps, and bounding boxes allow for semantically conditioned generation, while geometric modalities enable grounding the generation on 3D information. 4M's versatility in handling various modalities, its capacity to benefit from cross-training, and its ability to learn cross-modal predictive representations (as demonstrated in Sections 3 and 4) suggest its potential for extension to even more modalities.

**Pseudo labeled multimodal training dataset.** Training 4M models requires a large-scale and aligned multimodal/multitask dataset that contains all the above modalities/tasks and is sufficiently diverse. Most multimodal datasets, however, either do not contain all our pre-training modalities [32], are too small [94], or are not diverse enough [126]. For those reasons, we resort to pseudo labeling [42, 5] the publicly available Conceptual Captions 12M (CC12M) [19] as a binding dataset using powerful off-the-shelf models. Because this approach only requires access to a dataset of RGB images, it may scale to even larger web-scale image datasets [99, 14, 39].

**Tokenization.** All modalities are mapped to sets or sequences of discrete tokens (indices of a vocabulary) through the use of *modality-specific tokenizers*. Captions and bounding boxes are both treated as text and encoded using WordPiece [30]. For modeling bounding boxes, we follow the approach of Pix2Seq [21], that turns the task of object detection into a sequence prediction problem. RGB, depth, normals, semantic segmentation maps, and CLIP feature maps are tokenized using learned vector quantized autoencoders (VQ-VAE) [110]. Unlike Unified-IO [74] that represents all image-like modalities using an RGB pre-trained VQ-GAN [35], we instead use modality-specific tokenizers. This allows us to incorporate neural network feature maps that would otherwise be difficult to represent with existing image tokenizers. While mapping modalities to tokens and vice versa incurs a small computational overhead during inference, we avoid this overhead during pre-training by pre-computing the tokens while assembling our multimodal dataset.

We provide a detailed overview of the multimodal dataset, pseudo labeling procedure, and tokenization in Appendix B.

## 2.2 Multimodal Transformer

We design the architecture of 4M with efficiency, scalability, and simplicity in mind. 4M's architecture closely resembles a standard Transformer [112] encoder-decoder but includes a few crucial modifications to enable joint modeling of multiple different image-like modalities, such as RGB or semantic segmentation, but also of sequence modalities, such as captions or bounding boxes.

**Multimodal encoder.** The encoder is a standard Transformer encoder but features modality-specific learnable input embedding layers to map token indices to vectors. To each token of a specific modality, we add a learnable modality embedding and either 1D (for sequences) or 2D (for dense modalities) sine-cosine positional embeddings. To facilitate transfer learning, the encoder is additionally designed to accept RGB pixels using a learnable patch-wise linear projection, enabling it to double as a Vision Transformer [31] backbone.

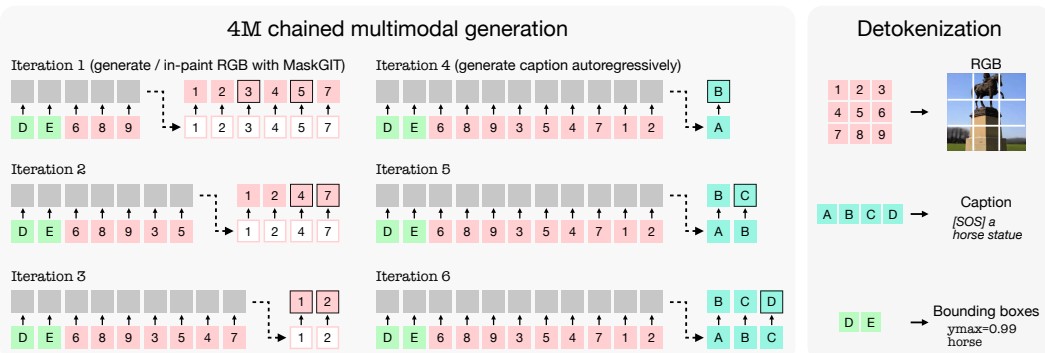

Figure 3: **Chained multimodal generation.** This simplified example illustrates the generation of a full RGB image from a partial RGB and bounding box input using the MaskGIT [17] decoding scheme, followed by autoregressive generation of a caption. Note that through chaining (i.e. using fully generated modalities as conditioning when generating subsequent modalities), we can predict multiple modalities in a self-consistent manner. This is in contrast to independently generating each modality from the original conditioning, where each generated output is consistent with the *input* but not necessarily with *other outputs*. See Figures 9 and 10 for visual examples of chained generation. Generated tokens can be turned back into images, text, and other modalities, using the detokenizers.

**Multimodal decoder.** The decoder handles tokens from both dense image-like and sequence-like modalities, with each type requiring a different approach. However, two aspects are common to all tokens: First, they can all freely attend to any encoder tokens in the cross-attention layers, ensuring full access to the encoded information. Second, we employ attention masks to separate decoder tokens of different modalities. This ensures that the decoder produces consistent outputs for each specific modality, irrespective of what other outputs are being generated simultaneously. For dense image-like modalities, the decoder input consists of mask tokens along with modality and positional information. The decoder's role is to predict this masked content. For sequence-like modalities, the input to the decoder comprises modality, positional, and content information. The decoder is tasked to predict the next token in the sequence. To ensure that each token is only influenced by preceding tokens (and not by any future tokens), we apply a causal mask to the self-attention, as is standard in autoregressive models. Since all target tasks consist of discrete tokens, we can use the cross-entropy loss *for all of them*, which we found removes the need for task-specific loss balancing and improves training stability. Further details on the architecture are provided in Appendix C.

## 2.3 Multimodal masking strategy

For multimodal pre-training, we use a pre-training strategy similar to MultiMAE [5], in that we sample and encode a small set of visible tokens/patches from all modalities, and train the model to perform cross-modal predictive coding.

**Input & target masking.** Dropping masked-out tokens and only encoding the small set of visible ones when performing masked image modeling has been shown to yield significant increases in training efficiency [48], and is crucial when training on multiple modalities [5]. The imbalance of the usually low number of input tokens and the much higher number of target tokens can induce significant computational costs in the decoder, even if they are small. We propose to use *target masking*, meaning that we do not decode all masked-out tokens, but only a randomly sampled subset. By fixing the number of randomly sampled input and target tokens (see Figure 2), 4M enables pre-training on many modalities while keeping training costs low. Similar to MultiMAE [5], we sample the number of input tokens per modality using a symmetric Dirichlet distribution with concentration parameter $\alpha$. We follow the same approach to also sample the number of target tokens per modality. After sampling the per-modality number of input and target tokens, we sample tokens from dense modalities uniformly at random, and perform span masking [86] on sequence modalities. The experimental consequences of these design choices are studied in Section 5, and in more detail in Appendix E.4.

## 3 Transfer Experiments

To assess the effectiveness of 4M as a pre-training strategy, we train two models: a base version 4M-B with 86M encoder parameters and a large version 4M-L with 303M encoder parameters (for more details, see Appendix C). We then transfer these trained models to several common downstream tasks and compare their performance against relevant baselines. To better control for the dataset, augmenta-

Table 1: **Transfer learning study:** We transfer 4M models to semantic and geometric downstream tasks and compare it to several baselines. For transfers to ImageNet-1K [29, 96], we first perform intermediate fine-tuning on ImageNet-21K [29, 93]. 4M outperforms the baselines on all tasks except for ImageNet-1K, surpassed by DeiT III which is a specialized model. In contrast to 4M, all of the baselines employed data augmentations to achieve their results. Best results per category are **bolded**.

| Method | Pre-training data | Data aug. | Extra labels | ImageNet-1K Top-1 acc. ↑ | COCO AP$^{box}$ ↑ | AP$^{mask}$ ↑ | ADE20K mIoU ↑ | NYU depth $\delta_1$ acc. ↑ |
|---|---|---|---|---|---|---|---|---|
| MAE B [48] | IN-1K | ✓ | ✗ | 84.2 | 48.3 | 41.6 | 46.1 | 89.1 |
| DeiT III B [108] | IN-21K | ✓ | ✗ | **85.4** | 46.1 | 38.5 | 49.0 | 87.4 |
| MultiMAE B [5] | IN-1K | ✓ | ✓ | 84.0 | 44.1 | 37.8 | 46.2 | 89.0 |
| 4M–B (RGB → RGB only) | CC12M | ✗ | ✗ | 82.8 | 42.3 | 36.6 | 38.3 | 80.4 |
| 4M–B (RGB → CLIP only) | CC12M | ✗ | ✓ | 83.4 | 46.6 | 39.9 | 43.0 | 85.7 |
| 4M–B | CC12M | ✗ | ✓ | 84.5 | **49.7** | **42.7** | **50.1** | **92.0** |
| MAE L [48] | IN-1K | ✓ | ✗ | 86.8 | 52.8 | 45.3 | 51.8 | 93.6 |
| DeiT III L [108] | IN-21K | ✓ | ✗ | **87.0** | 48.7 | 41.1 | 52.0 | 89.6 |
| 4M–L | CC12M | ✗ | ✓ | 86.6 | **53.7** | **46.4** | **53.4** | **94.4** |

tions, model architecture, and compute, which can significantly affect downstream performance, we additionally show self-baselines that are conceptually similar to MAE [48] (Masked RGB → RGB) and BEiT-v2 [82] (Masked RGB → CLIP).

The transfer tasks include ImageNet-1K classification [29, 96], COCO detection and instance segmentation [67], ADE20K semantic segmentation [131], and NYUv2 depth estimation [102]. While some transfer tasks have similarities to our pseudo labeled tasks, they are different instantiations (e.g., ADE20K instead of COCO semantic classes, or absolute depth instead of relative depth).

To make 4M models comparable to other ViT backbones, we train all methods by attaching transfer task-specific heads (e.g. Cascade Mask R-CNN [15, 47]) to the encoder, and discard any decoders. Note that we can also choose to keep the decoder for transfer learning, which we explore in Section 5. For comparability, we perform the transfers of all 4M and baseline models in the same controlled manner, by closely following commonly used settings from other papers. Exact training details are provided in Appendix D.

Results in Table 1 show that 4M transfers exceptionally well to all downstream tasks, outperforming the baselines on detection, segmentation and depth estimation. While on ImageNet-1K, 4M is outperformed by more specialized models such as DeiT III, the results demonstrate 4M to be a *versatile* vision model that can strongly benefit from being pre-trained on multiple (pseudo labeled) tasks. We note that preliminary experiments with an even larger 4M–XL model with 2.7B parameters showed overfitting to the pseudo labeled CC12M dataset, resulting in limited additional improvement on downstream tasks. While a larger dataset or adding data augmentations are therefore necessary to fully benefit from larger 4M models, we still observe significant improvements in generation quality and use the 4M–XL model in the next section.

## 4 Generative Capabilities & Probing the Learned Representation

4M can directly be used for generation of all pre-training modalities, through iteratively decoding tokens [17, 65, 18], as illustrated in Figure 3. In addition, we enable several other generative capabilities by utilizing two key aspects of 4M: The **first** is the fact that 4M is crucially able to generate *any* of the training modalities, either unconditionally or conditioned on *any* other set of modalities (see Figure 4 top). The **second** is the fact that 4M is trained using masking, which enables (conditional) in-painting and out-painting (see Figs. 1 and 4). Combining these two key aspects enables several *multimodal editing tasks*, such as *semantic editing*, *geometrically grounded generation*, or *guiding the generation with multiple strong and weak conditions* (via weighting).

To improve image generation fidelity, we trained a 4M–XL version and all subsequent images were generated with it. While 4M can be directly used for generation, there are certain common improvements we perform to improve image fidelity in the results shown. These include specializing 4M–L into a super-resolution variant that maps tokens of low-resolution generations to a higher resolution [18, 123], and specializing 4M models by fine-tuning them to be more aligned with specific generative use-cases (e.g. for text-to-image, or in-painting) [95]. See Appendix A for more details on these specializations.

**Probing learned representations through generation.** In addition to performing transfers, we can get a glimpse into what kind of (predictive) representations 4M learned by manipulating one part of

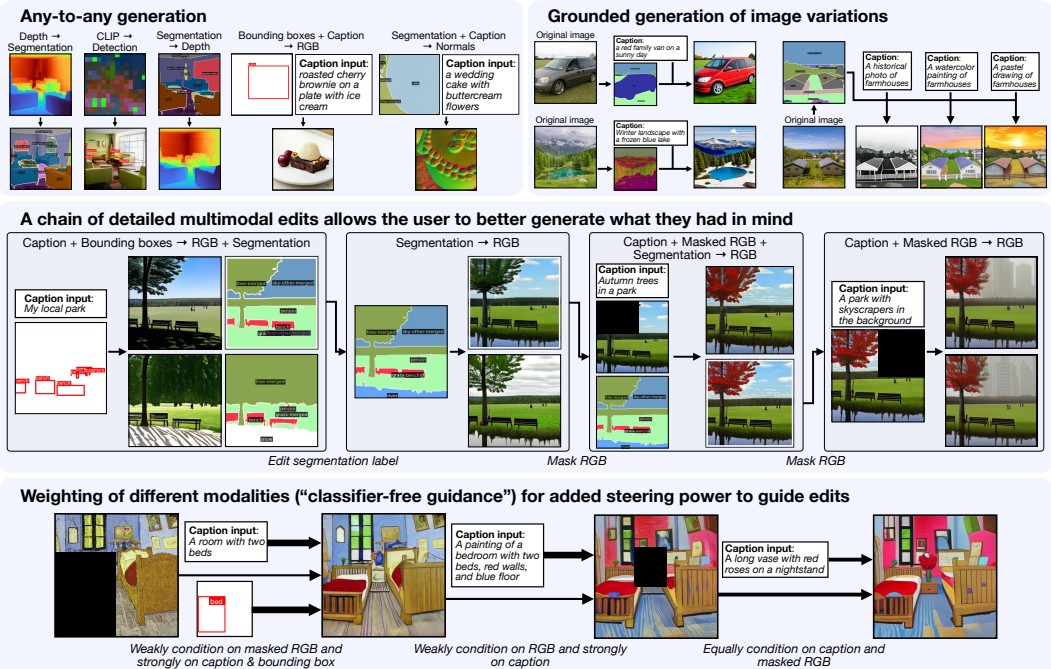

Figure 4: **Multimodal generation and editing.** 4M 's in-painting and any-to-any prediction abilities unlock a suite of multimodal generation and editing capabilities, which allow for fine-grained creative control. We show several key capabilities such as grounding the generation in predicted geometry, performing semantic edits, and being able to control how much certain input modalities influence the generation via weighting.

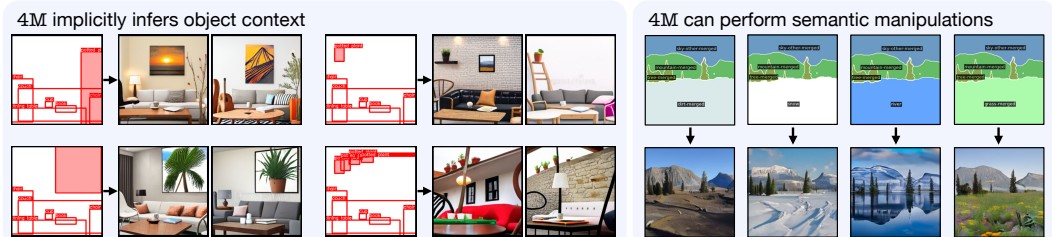

Figure 5: **Probing the learned representation through manipulation**: (Left): All of the conditioning bounding boxes are fixed except the marked "potted plant" ones that are changing. Depending on where the "potted plant" bounding boxes are placed, 4M infers how they can be part of the scene (e.g., as a painting or real plant) in a geometrically and physically plausible manner. (Right): Changing a single semantic class accordingly affects how 4M predicts the overall image. See more examples in Appendix A.4 and interactive visualizations on our website.

the input and keeping the remainder fixed [5]. Figure 5 shows a series of manipulations, in which 4M shows intriguing capabilities at predicting geometrically and physically plausible arrangements while taking into account the semantic context of the scene.

**Multimodal editing.** By combining the multimodal conditional generation and in-painting capabilities of 4M, we can perform various multimodal editing tasks, such as performing semantic edits or in-painting grounded by geometric conditioning (see Figure 4 top and middle). Drawing parallels to ControlNet [130], these allow for steering the generation using more than just text. However, 4M is able to perform these tasks with just a single network and condition on multiple (partial) modalities – *individually* or *simultaneously*. The conditions can either be hand-specified, or extracted from an image using 4M itself, thereby removing the need for specialist models to create the conditions.

**Multimodal weighted guidance.** Classifier-free guidance [50] has been shown to improve image fidelity in token-based generative models [40, 123, 18]. Inspired by Liu et al. [68] that perform compositional generation on multiple text conditions, we can guide our generation by weighting different (parts of) modalities by different continuous amounts – even negatively. This unlocks further

multimodal editing capabilities (see Figure 4 bottom), such as being able to weakly condition on certain modalities, or using negative weighing to avoid a certain concept in the generation. Multimodal guidance can be achieved by computing a weighted sum of the logits of an unconditional and each conditional case: $\text{logits}_{\text{guided}} = \text{logits}_{\text{uncond}} + \sum_{i=1}^{n} w_i \left( \text{logits}_{\text{cond, i}} - \text{logits}_{\text{uncond}} \right)$.

We provide additional visualizations covering 4M's wide range of generative capabilities on our website and in Appendix A.4.

## 5   Ablations

Pre-training on a large set of modalities using a masking objective creates a large design space that raise questions such as: what modalities should we pre-train on, what is the optimal masking ratio, or how do we select the number of tokens to mask from each modality? By performing a thorough ablation of these design parameters, we aim to find out which ones matter the most for multimodal pre-training and which ones do not. We used these findings to decide on the settings for the models shown in Section 3 and 4.

We first choose a *reference setting* and measure the deviation of the model performance as a result of the one aspect under study while keeping the rest fixed. Performance is measured through transferring the models to a large set of downstream tasks and measuring the validation set performance. The aim of the benchmarks is to measure how good a certain instantiation of 4M is at transferring both to *new target tasks*, but also to *unseen input modalities*. For that, we include tasks that use RGB (pixels) as inputs and transfer to a new target, such as COCO [67] object detection, ADE20K [131] semantic segmentation and ten transfers from RGB to dense tasks in Taskonomy [126] and Hypersim [94] (e.g. depth, curvature, segmentation). Furthermore, we include eleven single-modal and twelve multimodal transfers from some Taskonomy or Hypersim modalities to others (e.g. curvature → occlusion edges, or RGB + depth → segmentation).

For simplicity, and similar to pre-training, we model each of the benchmark tasks as predicting one set of tokens from another. This is achieved by training tokenizers on these modalities and tasks in the same way as we did for pre-training. All transfers are performed at 224 × 224 resolution. For comparability across modalities and tasks, we measure the validation set cross-entropy performance rather than task-specific metrics. Note that in the interest of space, we aggregate the results of the Taskonomy and Hypersim tasks, and denote them as *RGB→X*, *X→Y* and *X+Y→Z*. See Appendix E for further training details.

### 5.1   Reference model

Following common practice for the reference model size [30, 86], our model consists of 12 encoder and 12 decoder layers. We train it on CC12M [19] using all modalities as both inputs and targets, at resolution 224 × 224 pixels (corresponding to 14 × 14 tokens per dense modality), and using *no augmentations* such as cropping nor color augmentations. The total training length is fixed at 100B tokens, corresponding to roughly 400M masked samples. We set the number of randomly sampled input and target tokens to 12 each, and sample each using a symmetric Dirichlet distribution with parameter $\alpha = 0.2$. To determine the significance of various modeling choices, we adopt the approach used by Raffel et al. [86] that calculates the standard deviation of the transfer results of ten independently trained reference models.

### 5.2   Input modalities and target tasks

**Importance of target tasks for representation learning.** 4M is both a multimodal and a *multitask* training scheme and the choice of target task(s) is a powerful way of steering what representation the model learns [10, 109, 42, 107, 5]. To ablate this for 4M, we fix the input modalities to be either RGB or all the modalities (denoted as *"All"*), and vary the target tasks. The results in Table 2 mirror the findings of Sax et al. [98], that the optimal choice of pre-training setting depends highly on the type of transfer that is performed, and that there is no single pre-training task that performs best on all transfers. However, pre-training on all target modalities consistently outperforms the other single-task and multitask alternatives in terms of average loss, no matter what input modalities were used during pre-training. This makes it the preferred configuration for generalist models, especially when their future applications are unknown or varied.

**Importance of multimodal pre-training for transferring to new input modalities.** Table 2 shows that multimodal pre-training can significantly help with transferring to new input modalities (X→Y and X+Y→Z transfers), but comes at a performance loss at transfers that use RGB as the sole input modality. In Appendix E.4, we explore pre-training using mixtures of different masking strategies, which enables us to train models that perform well in both regimes.

Table 2: **Pre-training input and target modalities ablation:** The choice of pre-training tasks and modalities influences what representations the model learns, and how well it can be transferred to novel tasks and modalities. Here, *Geometric = RGB + Depth + Normals* and *Semantic = RGB + Segmentation + CLIP + Detection + Captions*. We show the average losses (↓) for several task categories and compute the average rank and best losses for *"RGB"* and *"All"* inputs separately. The reference model setting is indicated by ▷ and results that lie within two reference model standard deviations of the best result are **bolded**. Performing 4M pre-training on all input and target modalities is the most versatile choice, if the optimal set of pre-training modalities for any given downstream task is unknown.

| Pre-training inputs | Pre-training targets | COCO Det. | ADE20K Seg. | RGB→X | X→Y | X+Y→Z | Avg. Loss | Avg. Rank |
|---|---|---|---|---|---|---|---|---|
| RGB | RGB | 3.14 | 6.21 | 5.03 | 6.94 | 6.15 | 5.49 | 8.00 |
| RGB | Depth | 3.11 | 6.06 | 4.72 | 6.89 | **5.84** | 5.32 | 4.00 |
| RGB | Normals | 3.12 | 6.02 | **4.66** | **6.83** | 5.87 | **5.30** | 3.20 |
| RGB | Segmentation | 3.17 | **5.94** | 4.84 | **6.86** | 5.89 | 5.34 | 4.60 |
| RGB | CLIP | 3.07 | 6.11 | 4.83 | **6.85** | 5.94 | 5.36 | 4.80 |
| RGB | Detection | **2.78** | 6.11 | 5.03 | 7.07 | 6.24 | 5.45 | 7.20 |
| RGB | Captions | 3.45 | 6.55 | 5.92 | 7.35 | 6.86 | 6.03 | 10.00 |
| RGB | Geometric | 3.11 | 6.08 | **4.70** | 6.88 | **5.85** | 5.32 | 3.80 |
| RGB | Semantic | 2.88 | 5.99 | 4.86 | 6.97 | 6.06 | 5.35 | 5.40 |
| RGB | All | 2.90 | 5.99 | 4.74 | 6.91 | 5.93 | **5.29** | 4.00 |
| All | RGB | 3.21 | 6.20 | 5.07 | **6.75** | 5.85 | 5.42 | 3.80 |
| All | CLIP | 3.19 | 6.18 | 5.06 | 6.80 | 5.88 | 5.42 | 3.80 |
| All | Geometric | 3.20 | **6.13** | **4.98** | **6.72** | **5.76** | **5.36** | 1.80 |
| All | Semantic | **3.05** | 6.13 | 5.16 | 6.77 | 5.87 | 5.39 | 3.40 |
| ▷ All | All | **3.06** | **6.11** | 5.07 | **6.75** | 5.80 | **5.36** | 2.20 |

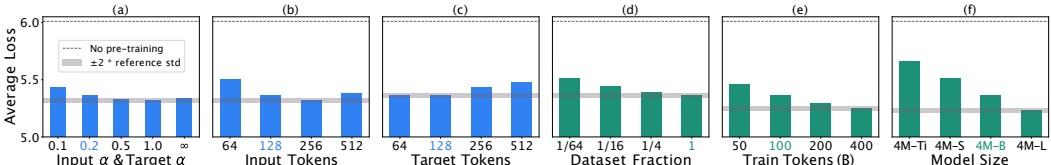

Figure 6: **Ablations results:** We ablate several key design choices of the multimodal masking objective (in blue), and study how well 4M scales (in green). We show the overall average losses (↓) and highlight the reference model setting in blue / green. A detailed breakdown of the task losses is provided in Appendix E.4 and Appendix E.5.

## 5.3  Multimodal masking strategy

Multimodal masking is at the core of 4M, so in this section, we ablate how modality tokens should be sampled, and how many tokens we should encode and decode.

**Modality proportions in masking.** We ablate various choices of Dirichlet parameter $\alpha$, both for the input and target sampling. If $\alpha$ is low, the sampling procedure will often select cases where most of the tokens are sampled from only one modality. If $\alpha$ is high, however, most samples will contain tokens from all modalities to equal proportions. Results in Figure 6 (a) show that uniformly sampling over the simplex performs best on average, but not by a large margin.

**Input masking budget.** The difficulty of the multimodal masked modeling task is largely determined by the number of visible (non-masked) input tokens. Encoding only the small set of visible tokens can significantly improve training efficiency [48]. Figure 6 (b) shows that with a fixed training token budget, training with 128-256 input tokens performs well.

**Target masking budget.** We can decide to decode merely a small random subset of all remaining masked-out tokens, which is especially important for enabling efficient multimodal training with large decoders. As Figure 6 (c) shows, decoding only a small random subset of all targets performs well (for a fixed number of total training tokens), while also reducing computational costs.

## 5.4  How well does 4M scale?

Scalability is a key property that models and training objectives should have. We therefore ablate the following three axes: To ablate the *dataset size*, we train 4M models on various subsets of CC12M, down to 1/64th of the full dataset. To ablate the *training length*, we vary the total number of tokens seen. We define *tokens seen* as the total number of both input tokens and target tokens the model was

trained on. To ablate the *model size*, we train different sizes of 4M models, ranging from Tiny (`4M-Ti`) with 24M parameters to Large variants (`4M-L`) with 705M parameters. Exact model specifications are given in Appendix C.1. Figure 6 (d), (e), (f) show that 4M scales with dataset size, training length, and model size, respectively.

For additional ablations on the architectural and training design choices of 4M, see Appendix E.

## 6   Related Work

Large language models have been demonstrated to be capable of performing a diverse range of tasks out of the box [86, 12, 81, 25, 45] by training on large datasets with simple objectives [30, 84, 86, 106]. In vision, however, many scaling efforts have instead focused on training specialized models on a single task and modality, such as predicting masked RGB pixels [20, 31, 4, 48, 118, 34], discrete tokens [7, 132], or other (deep) features [117, 6, 82, 114, 36, 70] from RGB inputs. Training models instead on multiple tasks [16, 33, 63, 42, 92, 11] and modalities [85, 122, 76, 59, 133, 1, 3, 57, 43], or both [54, 103, 114, 5, 44] usually requires modality-specific modeling choices, making it difficult to extend these methods.

While some recent works aim to consolidate various modalities and tasks by representing them as images [77, 8, 115], these approaches have limitations when dealing with modalities that cannot be readily converted into images, such as text or neural network feature maps. Instead, 4M adopts the approach of Pix2Seq [21, 22] and Unified-IO [74] which addresses these issues by unifying the representation space on which models are trained through tokenization [110, 35, 30]. However, unlike methods like Unified-IO which operate on a single RGB image tokenizer, 4M's ability to work with multiple modality-specific tokenizers enables scaling to visual modalities beyond those that can be represented as images, such as neural network feature maps. 4M also builds upon the multimodal masking approach of MultiMAE [5] and extends it beyond image-like modalities.

Both token-based generative models [88, 123, 17, 18, 65] and diffusion models [89, 79, 95, 97] have been mostly limited text-to-image generation. While there are works that enable a greater amount of control by conditioning on additional modalities, they are either very limited in the number of ways they can be conditioned on [58, 40, 113, 119, 62, 9, 128], or require training a separate model for each new modality [130]. 4M flexibly allows for conditioning on any subset of the training modalities, and can, likewise, generate all these modalities, unlocking powerful generative editing capabilities.

## 7   Conclusion and Limitations

4M is a generalist framework for training multimodal and multitask models that not only perform many key vision tasks out of the box, but also demonstrate strong transfer results to a wide range of downstream tasks. 4M's in-painting and any-to-any generation capabilities enable it to perform a wide range of multimodal generative and expressive editing tasks – all using a single model. In the following, we discuss some limitations of our approach and potential future work.

*Additional modalities.* While 4M already includes a number of modalities, from semantics-based and geometry-based to text-based, bringing additional modalities could vastly improve its usability. For example, training on features extracted from a large language model has been shown to significantly boost text-to-image generation capabilities [97, 123, 18]. Introducing modalities like edges, sketches, or human poses has the potential to greatly improve the expressiveness [130] of 4M, but it may also be expanded to videos or multi-view imagery to unlock spatial-temporal generation and editing capabilities. We anticipate 4M to conveniently extend to such new modalities.

*Tokenizer quality.* 4M can benefit from better tokenizers, both in terms of generation and transfer results, but there are limits to the amount of information that can be encoded in tokenized patches. Operating on tokens that cover a smaller image region, or operating on higher-resolution images may improve image quality, but is expected to be computationally more expensive. Generally, improvements along this direction are expected to directly boost the performance of 4M.

*Dataset size and quality.* While our binding dataset choice of CC12M is standard, 4M can benefit from training on significantly larger datasets [99, 14, 39]. Web-scraped image-text datasets contain many low-quality images, as well as captions that are not related to the image content. Fine-tuning 4M on a more curated dataset like LAION-Aesthetics V2 [99], or tuning the model using reinforcement learning [83] could significantly improve generation quality and diversity. We leave this for the future.

**Acknowledgements.** We thank Hanlin Goh and Elmira Amirloo Abolfathi for their valuable feedback on earlier versions of this manuscript.

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

# Appendix

# A    Generative Capabilities & Probing the Learned Representation

## A.1    Token super-resolution

For generating high-resolution images, it is common to first generate images at a lower resolution, followed by a super resolution step [52, 18, 123]. These two-stage approaches often use a large model to generate a low-resolution image that contains most of the semantically and geometrically important information, followed by one or more smaller super-resolution models that fill in higher resolution details. Directly generating high-resolution images with Transformers is both computationally expensive due to the long sequence lengths, but has also been shown to perform worse at generating semantically coherent images [18] compared to cascade approaches. 4M is trained at a base resolution of $224 \times 224$ pixels, which corresponds to $14 \times 14$ tokens. At this low image resolution, fine details are difficult to model. We therefore train a super-resolution model, 4M-SR, starting with 4M-L as the base and fine-tune it to map from low-resolution image tokens to higher-resolution tokens. All images shown are generated at the base resolution using 4M-XL, and subsequently up-sampled to double the resolution using 4M-SR.

**Training details.** For the high-resolution tokens, each $448 \times 448$ pixel image is represented by $28 \times 28$ tokens. The super resolution training-scheme is very similar to the base resolution 4M pre-training scheme, but the inputs to the model consist of a random subset of low-resolution and high-resolution tokens of all modalities, while the target is a random subset of all high-resolution tokens. At inference-time, one or multiple full low-resolution modalities (and sequence-like modalities) are used as conditioning to decode the high-resolution targets step-by-step.

To train 4M-SR, we start from the pre-trained 4M-L and continue training it for an additional 100B tokens (20% of the pre-training length). For each sample, we randomly sample the input budget uniformly between 64 and 1024 to better accommodate for the varying number of tokens throughout the generation process, and keep the target budget fixed at 1024. As full captions are otherwise rarely seen during training, we train on a mixture of masking strategies where 1/4 of the input samples are heavily skewed towards fully unmasked captions ($\alpha = 5.0$ for captions with $p_{\text{mask}} = 0$, and $\alpha = 0.05$ for all other modalities).

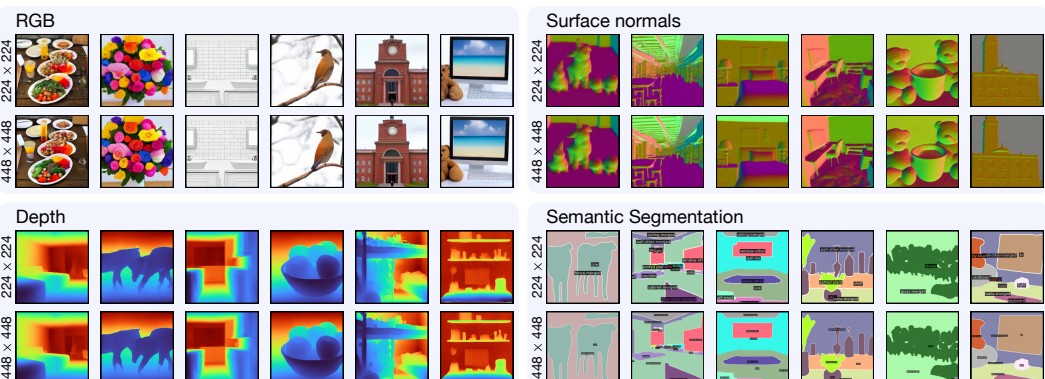

Figure 7: **Token super-resolution:** We specialize a version of 4M-L for performing super-resolution in the space of tokens ($14^2 \rightarrow 28^2$ tokens) to double the resolution ($224^2 \rightarrow 448^2$ pixels) and in the process add more details to the generated images. Just like the base model, it can generate any high-resolution modality conditioned on any low-resolution modality, but we use it to super-resolve modalities we generated at the base resolution. Best viewed when zoomed in.

## A.2    Generation-specific specializations of 4M

Adapting 4M to specific use-cases by fine-tuning it either on a subset of pre-training modalities, or fine-tuning it with a different token sampling distribution can improve performance on select downstream tasks. In a similar manner, to optimize a 4M model for text-to-image generation, we can fine-tune it by skewing the token sampling distribution to include more captions as input. To train this caption-skewed model, we start from a pre-trained 4M model and continue training it for an additional 50B tokens (10% of the pre-training length). For each sample in a batch, we randomly sample the input budget uniformly between 64 and 256 to better accommodate for the varying number of tokens throughout the generation process, and keep the target budget fixed at 256. To skew the model towards full captions, we train on a mixture of masking strategies where 1/3 of the input

samples are heavily skewed towards fully unmasked captions ($\alpha = 5.0$ for captions with $p_{\text{mask}} = 0$, and $\alpha = 0.05$ for all other modalities).

All text-to-image results in this section were generated using a caption-skewed 4M specialization.

## A.3    Generation procedure details

Token-based masked image models that were trained with variable masking rates can be directly used for image generation [17, 65, 18]. These models can be seen as order-agnostic autoregressive models [53] in which generation is performed by iteratively decoding tokens in any order. Unlike traditional autoregressive models which need to generate tokens one-by-one in a pre-determined order, these models are able to speed up inference by parallelizing the decoding process because the distribution over each masked token can be predicted at the same time.

**Generation schedule.** The generation process is guided by a generation schedule that outlines a sequence of generation steps. Each step in the sequence determines the target modality, the decoding scheme (described below), the number of tokens to decode, the temperature, and the top-k and top-p sampling values.

**Decoding schemes.** A crucial feature of our method is the flexibility to incorporate different decoding schemes, such as for different target modalities. We describe these decoding schemes below:

- **MaskGIT [17] (for image-like modalities).** This parallel decoding scheme generates the entire image in a pre-determined number of steps (see Figure 3). At every prediction step, we encode all visible tokens and decode all masked out tokens. We then sample from their predicted distributions and choose the $n$ most confident tokens, where $n$ is determined by the generation schedule. Finally, we add the $n$ predicted tokens to the input and repeat the previous steps.
- **Random order autoregressive (ROAR) (for image-like modalities).** This scheme is conceptually similar to the decoding scheme proposed in XLNet [120]. Unlike MaskGIT, we do not decode all masked-out tokens at every step, but instead randomly select $n$ tokens to decode.
- **Left-to-right autoregressive (for sequences).** Besides images, we are able to generate sequence modalities such as captions and bounding boxes by autoregressively decoding them with the Transformer decoder of 4M (see Figure 3).

**Chained generation.** Because any generated modality can be used as a conditioning too, we can perform *chained generation* of several modalities, one after another, with each fully generated one being added to the conditioning of the next (see Figure 3). Performing generation in this chained manner results in each additional modality being generated in a consistent manner, as shown in Figures 9 and 10. In addition, we found that for certain generative tasks, such as caption-to-RGB, generating intermediate modalities such as CLIP tokens can further improve image fidelity (see Figure 9).

**Classifier-free guidance.** Classifier-free guidance is crucial for improving both image fidelity and how well the generation matches the conditioning. It is most commonly used in diffusion models [50], but can be applied in token-based models as well [40, 123, 18]. We perform classifier-free guidance by computing a weighted combinations of the logits of a forward pass with the conditioning and one without the conditioning:

$$\text{logits}_{\text{guided}} = \text{logits}_{\text{uncond}} + w \left( \text{logits}_{\text{cond}} - \text{logits}_{\text{uncond}} \right).$$

Here, $w$ is the guidance scale. When performing chained generation, we add each fully generated modality to the set of guided modalities.

**Multimodal guidance.** While guidance has been shown to significantly improve image quality, it can still happen that generative models ignore parts of the input, unpredictably focus on some parts more than others, or generate undesired concepts. Negative prompting [78] is a popular way of keeping the model from generating undesired concepts. Liu et al. [68] show that performing compositional guidance on multiple conditions can further improve text-image similarity. In a similar way, we can perform compositional generation by weighting different (parts of) modalities by different continous amounts – even negatively. We can do this by computing a weighted sum of the logits of an unconditional case and the logits of each conditional case:

$$\text{logits}_{\text{guided}} = \text{logits}_{\text{uncond}} + \sum_{i=1}^{n} w_i \left( \text{logits}_{\text{cond, }i} - \text{logits}_{\text{uncond}} \right).$$

Here, $w_i$ are the guidance scales for the different conditions. For example, this allows 4M to generate semantically or geometrically similar variants of images by weakly conditioning on their extracted segmentation, normal, or depth maps (see Figure 13). It can further be used for fine-grained steerability of multimodal edits (see Figs. 16, 14, and 15), or to use negative weighting to avoid certain concepts from being generated (see Figure 17).

### A.4 Additional visualizations

**RGB→X.** The model can solve several common RGB → X vision tasks out of the box, such as predicting normals, depth, semantic segmentation maps, or performing object detection and captioning. In Figure 8, we show examples of this functionality.

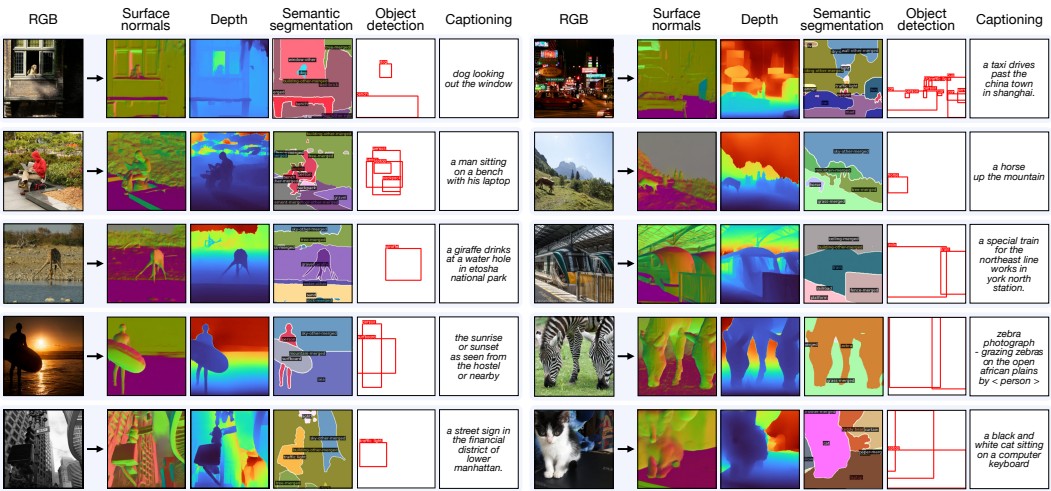

Figure 8: **RGB→X:** 4M can perform several common vision tasks, such as predicting surface normals, depth, semantic segmentation maps, or performing object detection and captioning.

**Chained generation.** For certain generation tasks, like text-to-image, we observed that generating intermediate modalities, such as CLIP, before generating the RGB image can improve image fidelity. This leads to a form of *progressive self-conditioning*, which our model particularly enables as it includes several modalities. We demonstrate this phenomenon in Figure 9.

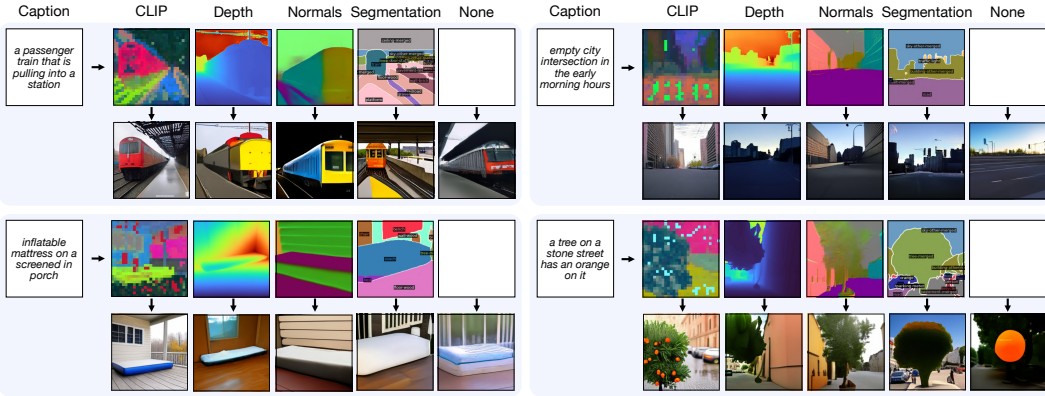

Figure 9: **Chained generation:** First, we generate intermediate modalities from the caption (CLIP, depth, etc.). Then we generate the RGB images by conditioning on both the caption and the intermediate modality, i.e., caption → intermediate modality → RGB. We compare this to directly predicting the RGB image from the caption, i.e., caption → RGB, in the 'none' column. We observe that certain intermediate modalities like CLIP can improve image fidelity. As 4M includes multiple modalities, it particularly enables this form of progressive self-conditioning and engineering a sequence of modalities for a better final generation.

**Any-to-any generative modeling.** 4M can be used to generate any modality from any other (and partial) subset of modalities. Figure 10 displays examples where all modalities are generated from a single input, and Figure 11 shows several examples of generating different modalities using two full inputs. Subsequent figures will make use of this functionality and show masked predictions and examples with 1-3 different conditionings.

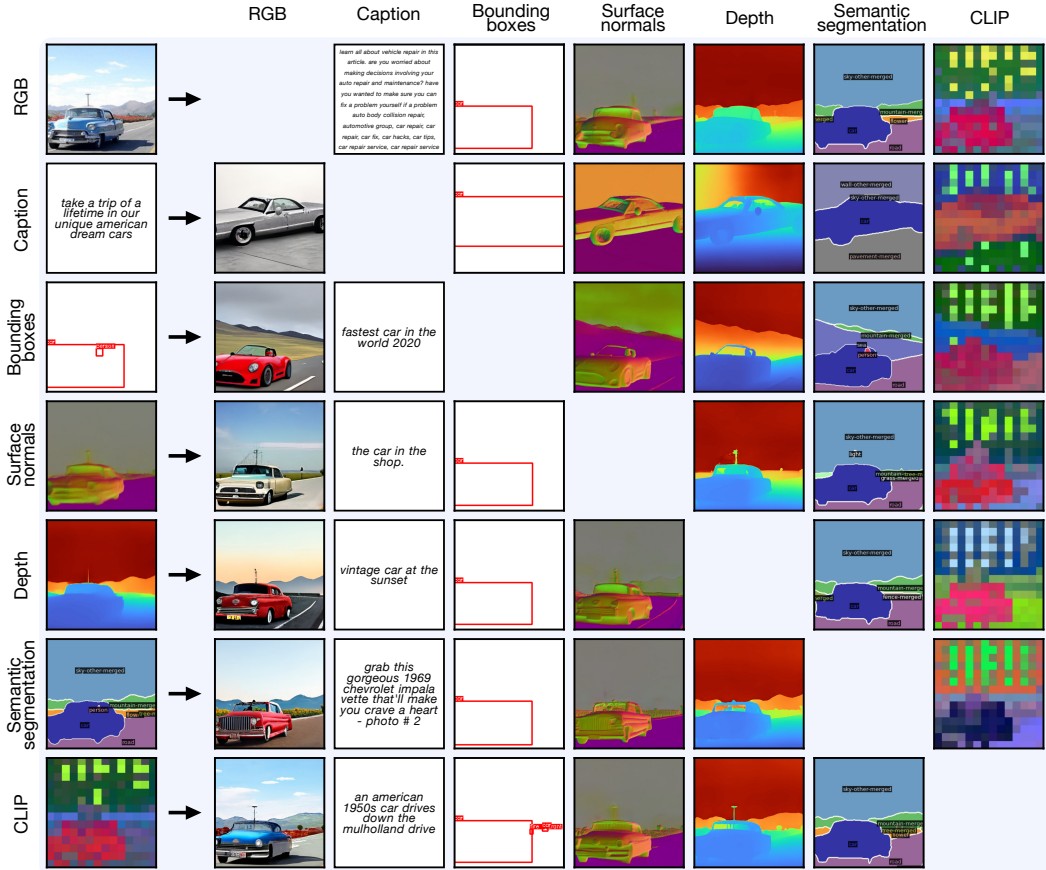

Figure 10: **One-to-all generation:** Starting from a single input modality, we use 4M to generate the corresponding outputs for all others. Each row shows the outputs from a different starting modality, all from the same original sample. Chained generation is used to ensure consistent outputs.

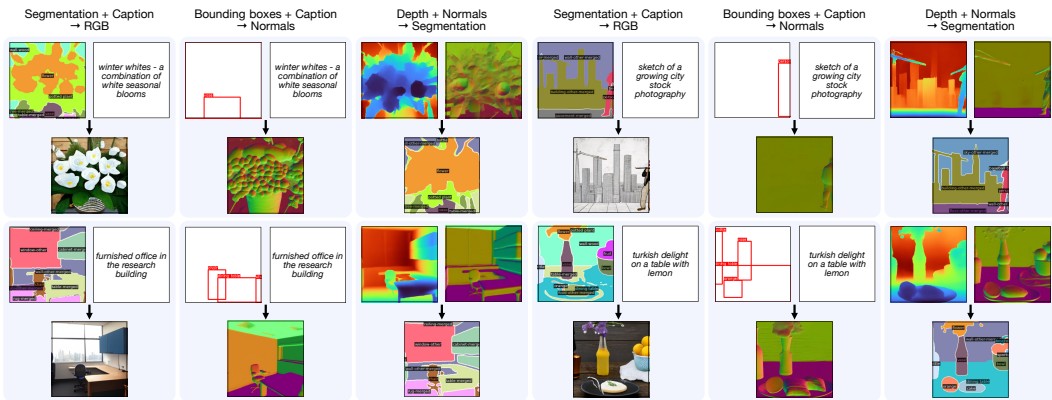

Figure 11: **Any-to-any generation:** 4M can perform *any-to-any* generation, meaning that any modality can be generated by conditioning on any subset of modalities.

**Conditional in-painting.** By virtue of being trained on a masking objective, 4M can be used for conditional and unconditional in-painting (see Figure 12). In-painting coupled with 4M 's ability to predict any modality from any other and flexible multimodal conditioning is the basis for several multimodal editing capabilities (see Figure 4 middle and bottom).

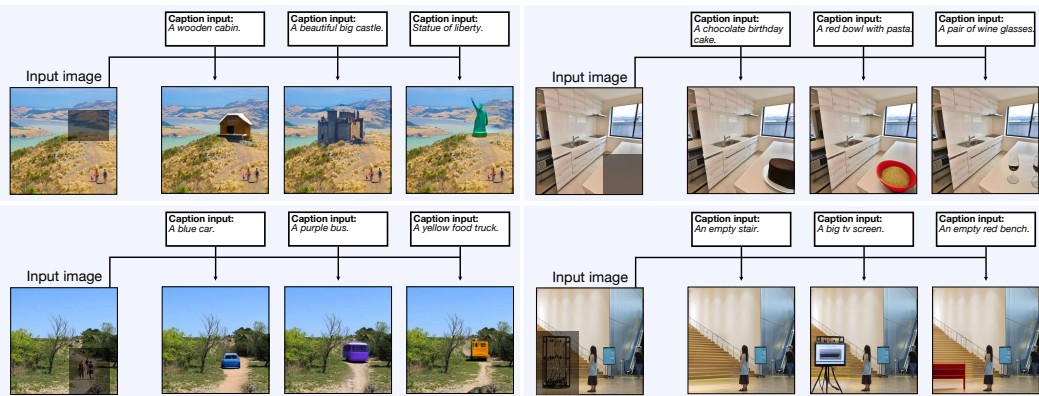

Figure 12: **Conditional in-painting:** 4M can be used for conditional in-painting by masking out a selected image region and generating the remaining tokens using the masked image and a caption (or other modalities) as conditions.

**Grounded generation of image variations.** 4M can be used to generate different versions of images by grounding the generation in some aspect extracted from a reference input image (see Figure 13). For example, we can take an image and generate variations that respect the geometry of the original image but may contain completely different colors by conditioning only on extracted surface normals. This is conceptually similar to ControlNet [130], but 4M is able to perform this using a *single model* trained in *one single run*, and 4M is able to flexibly use any combination of input modalities (see Figure 11). In addition, unlike ControlNet which requires external networks to extract the conditioning modalities, 4M models are able to generate all of them by themselves.

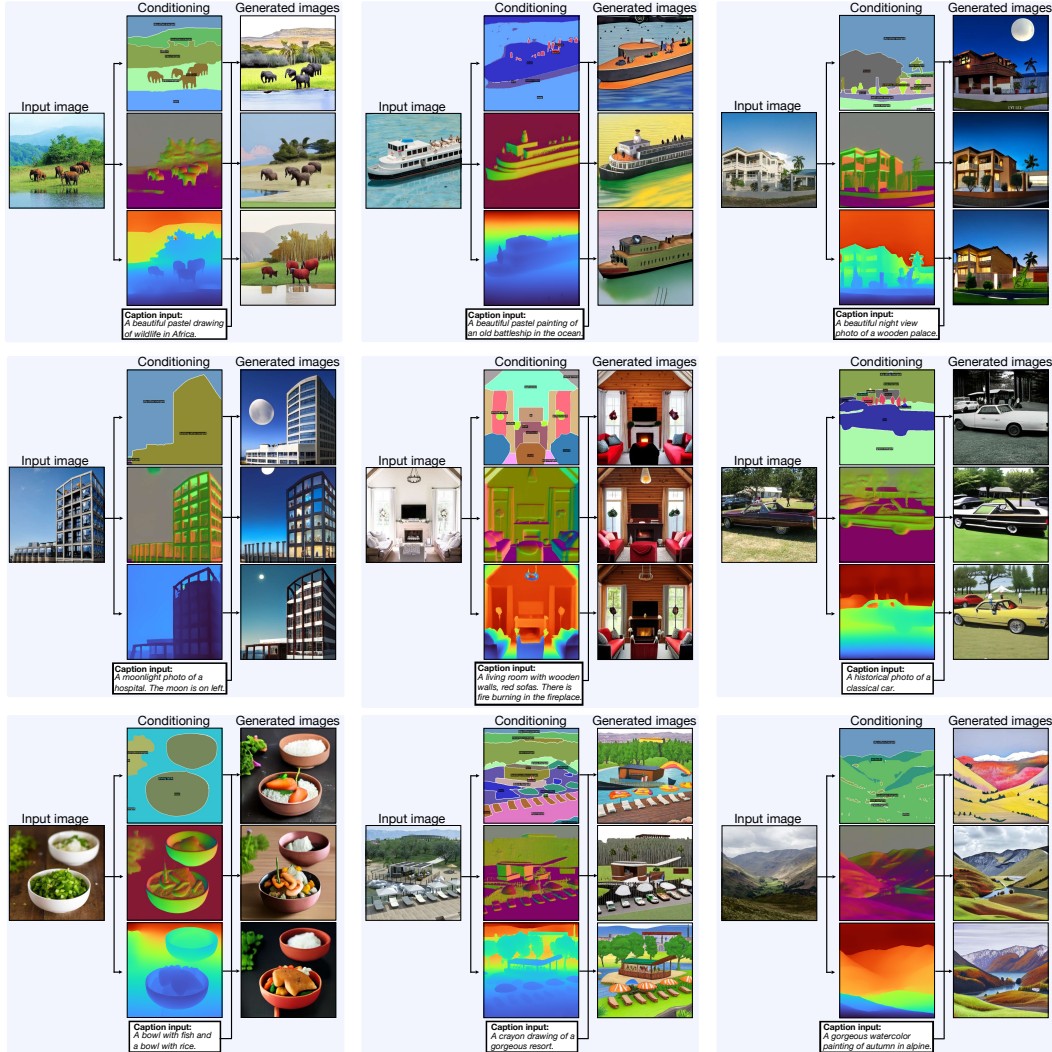

Figure 13: **Grounded generation of image variations:** 4M can be used to generate variations of a reference image by conditioning on a modality extracted from the original image. This allows for generation of diverse images that are similar geometrically, semantically, or in another (user-specified) aspect.

**Multimodal guidance.** Multimodal guidance, as explained in Appendix A.3, can be used for various fine-grained editing tasks. We show here several demonstrations of how it can be used to (1) use *weak/strong weighting* to generate images that loosely/strongly match a certain input modality, and (2) avoid the generation of certain undesired concepts via *negative weighting*.

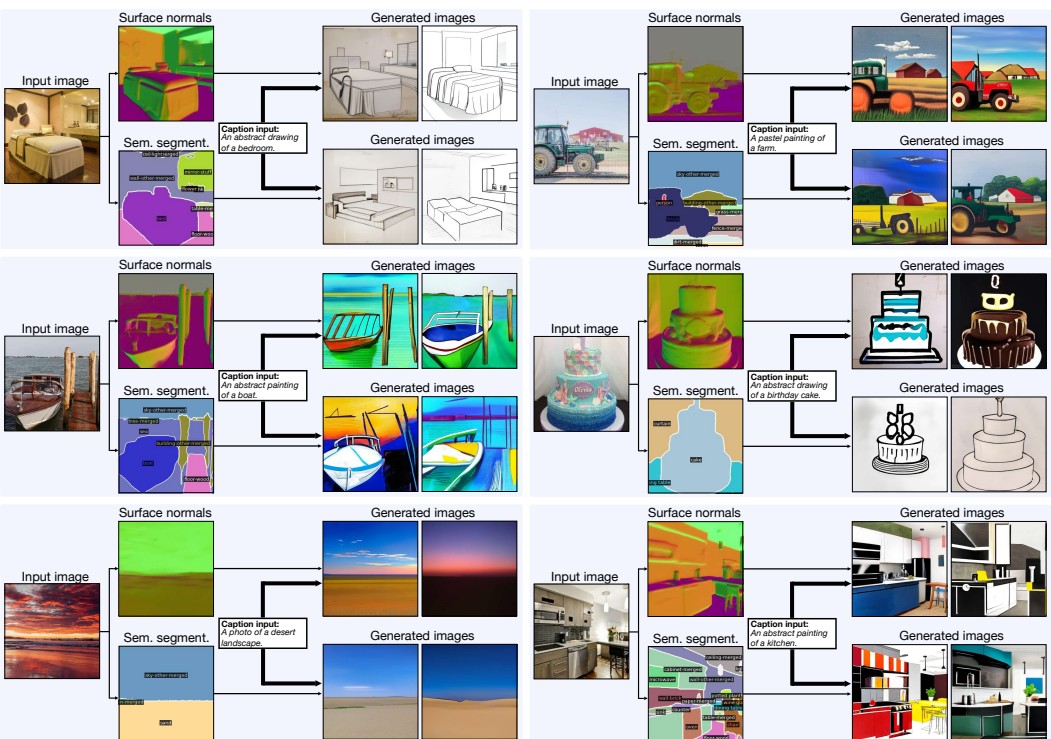

Figure 14: **Multimodal guidance:** By weakly conditioning on geometry (here surface normals) or semantics (here semantic segmentation maps), and strongly conditioning on a caption, we can generate image variations that are only loosely grounded in some aspect of the original image.

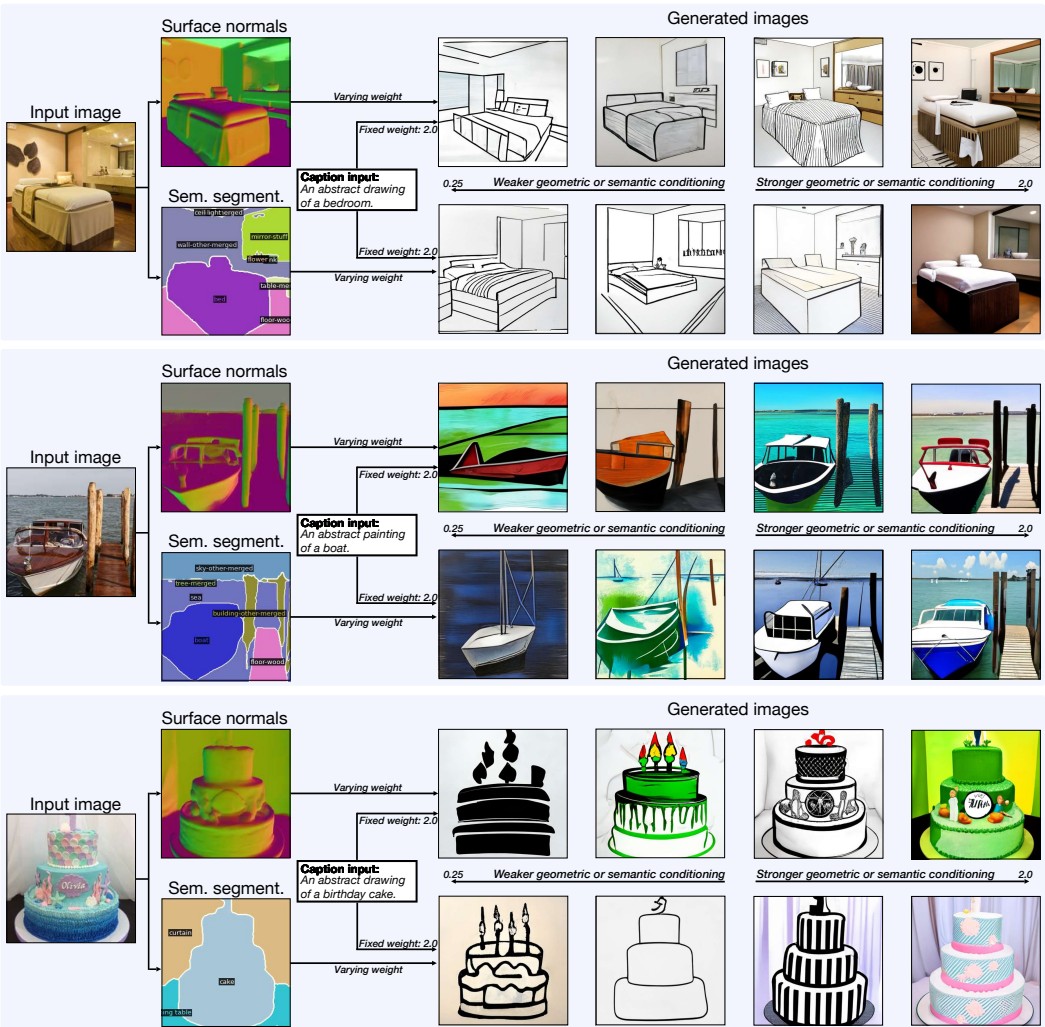

Figure 15: **Multimodal guidance:** Similar to Figure 14, we show here multimodal guidance with captions and either a normal or semantic segmentation map. We keep the weight of the caption fixed and vary the weight of the other modality. The examples demonstrate how a user can effectively control to which degree the generation should use a certain conditioning.

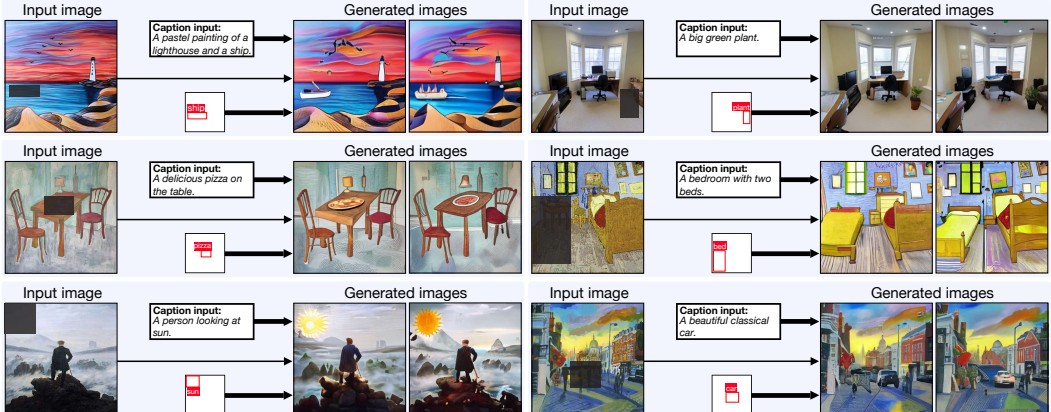

Figure 16: **Multimodal guidance:** In this example, we weakly condition on a masked RGB image, and strongly on a caption and bounding box. The generations follow the caption and bounding box precisely, while showing some degree of variation for the rest of the generated image.

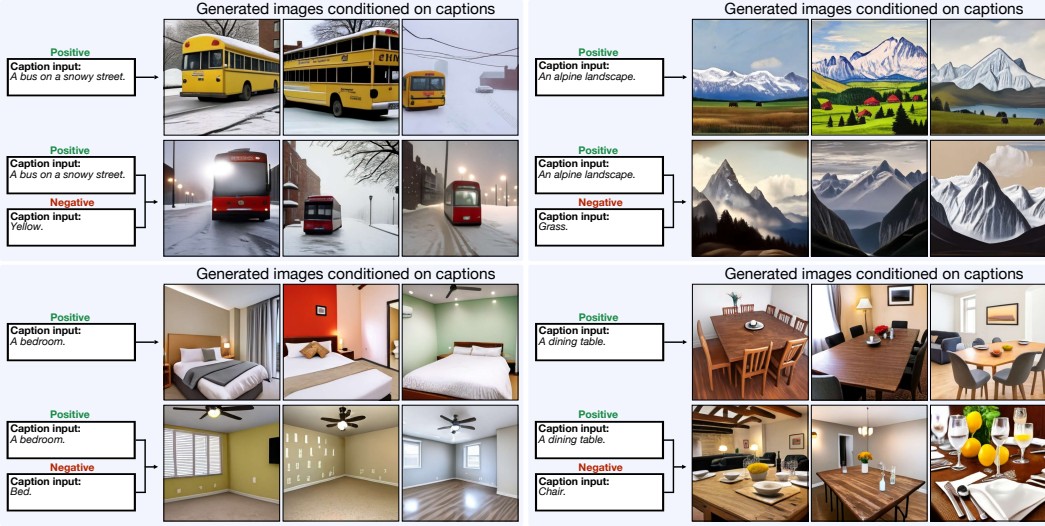

Figure 17: **Multimodal guidance: Multi-captioning with *negative weighting*.** (Top of each figure): Standard text-to-image generation can create pleasing results, but the user might want greater control over the generation to counteract dataset biases (e.g., most generated buses are yellow), or may have envisioned a different image. (Bottom of each figure): Enabled by the process described earlier, by adding an additional caption with a *negative* guidance weight, we can steer which concepts are generated, and which are not.

**Probing the learned representation.** We can probe what representations 4M models learned either by performing transfers to various downstream tasks or manipulating parts of the input and observe how 4M predicts some readout modality. Concretely, we perform these representation probes by keeping the entire input fixed and manipulating only a single aspect of it.

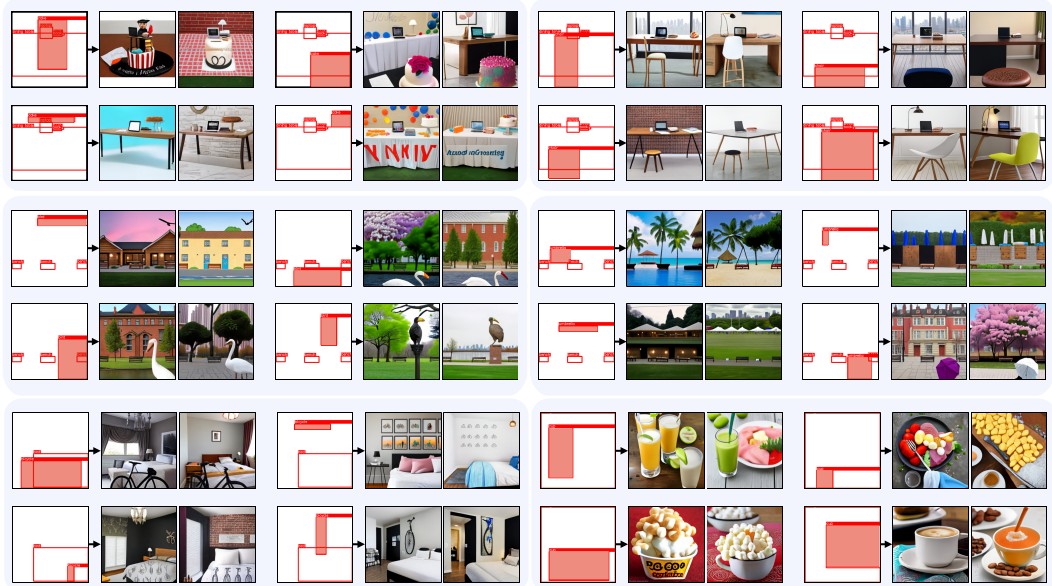

Figure 18: **Probing 4M representations:** In this example, we condition the generation on several bounding boxes. For each of the generations inside the blue cells, we keep all bounding boxes fixed, *except for one* (marked in red shade), which we move around and resize. Depending on where the bounding box is placed, 4M implicitly infers the semantic and geometric properties of the object. For example, placing a bicycle bounding box in front of a bed bounding box generates a real bicycle, while placing it above generates pictures of bicycles.

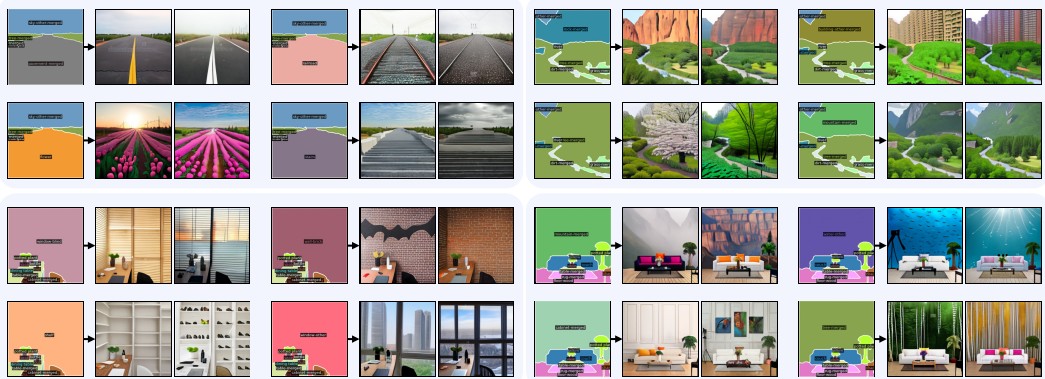

Figure 19: **Probing 4M representations:** Here, we condition the generation on segmentation maps in which we manipulate a single segment and keep all others fixed. 4M creates plausible generations, no matter what class the segment was changed to.

# B  Multimodal Dataset & Tokenization

We pseudo label a large aligned multimodal dataset using CC12M [19] and several pre-trained networks to map RGB images to pseudo labels. To enable training a single Transformer on a large number of vastly different modalities, we resort to tokenization. Concretely, we train modality-specific tokenizers, allowing us to map different modalities to sequences or sets of tokens.

## B.1  Pseudo labeled multimodal training dataset details

We trained all 4M models using an aligned multimodal dataset that we created by pseudo labeling CC12M. CC12M already contains aligned RGB images and captions, and we pseudo labeled the remaining tasks using powerful off-the-shelf models.

**Surface normals & depth.** To introduce geometric priors and to enable generation and transfer learning conditioned on measured or estimated geometric modalities, we pseudo labeled both surface normals and scene depth. We used DPT-Hybrid [91] models trained on Omnidata [32] using cross-task consistency [125] and 3D data augmentations [61] to predict normals and depth from the CC12M RGB images. To encode images using DPT, we resized their heights and widths to the closest multiple of 32, while keeping the maximum side length below 768 to avoid prediction degradation from too large image sizes. We found that even though the Omnidata DPT models were mostly trained on simulated and indoor scenes, they generalize well to a wide range of scenes.

**Semantic segmentation.** For introducing dense semantic priors, semantic segmentation maps can be useful as it allow for fine-grained creative control when conditioned during generation [40]. We pseudo labeled semantic segmentation labels using a Mask2Former [24] with a SwinB [71] backbone trained on COCO [67] panoptic segmentation. We select the labels by taking the `argmax` over the predicted semantic classes.

**Bounding boxes.** To add object bounding boxes to the training data, we use a ViTDet ViT-H model [66] initialized from MAE weights [48] and fine-tuned on COCO [67]. We filter the detected bounding boxes by removing all instances with a confidence score below 0.6.

**CLIP feature maps.** Tokenized CLIP [85] feature maps have been demonstrated to be powerful targets for masked image models [82, 37, 36]. We use the ViT-B/16 visual backbone of a CLIP-B16 model to extract dense feature maps from its last Transformer layer. To visualize a $H \times W \times D$ dimensional CLIP feature map, we project all $H * W$ entries onto the first three principal components computed from said feature map, interpreting the normalized values as R, G, and B colors [126].

## B.2  Tokenization of captions & bounding boxes

We follow Pix2Seq [21, 22] and treat detection as a sequence prediction problem. For example, a scene containing a bounding box around a cat and a plant could be parameterized as `xmin=0.15 ymin=0.3 xmax=0.65 ymax=0.5 cat xmin=0.75 ymin=0.3 xmax=0.95 ymax=0.8 potted plant [EOS]`. In practice, we model the corner coordinates (i.e. the minimum and maximum $x$ and $y$ coordinates) using a resolution of 1000 special tokens per each of those cases – namely we represent bounding boxes using `xmin=0, ..., xmin=999, ymin=0, ..., ymin=999, xmax=0, ..., xmax=999, ymax=0, ..., ymax=999`. In contrast to Pix2Seq, our method involves masking parts of the object sequence, which requires two modifications to the sequence construction pipeline to make this task more manageable: First, we order the objects in the sequence by their distance to the origin. Second, unlike in Pix2Seq where corner coordinates share tokens, we assign separate tokens for each of the corner coordinates, resulting in a vocabulary size 4 times larger than in Pix2Seq. Assigning separate tokens for each corner coordinate ensures that the meaning of each token is not made ambiguous when surrounding coordinates are masked out.

We jointly fit a WordPiece [30] text tokenizer on all captions and the 4000 special bounding box tokens. COCO class labels for the bounding boxes are treated as special tokens, meaning that if they appear in any caption, they get mapped to the same tokens. The joint text and bounding box vocabulary has a size of 30K.

## B.3  Tokenization of dense modalities

We tokenize all dense modalities using variants of vector-quantized autoencoders (VQ-VAEs) [110]. To avoid blurry and unrealistic images, we train the tokenizers for RGB, normals and depth using diffusion decoders. All tokenizers are first trained on 100 epochs of an ImageNet-1K version of the pseudo labeled dataset (see Appendix B.1), after which they continued training for 15 epochs on the

pseudo labeled CC12M dataset. We follow up the training at the base resolution of $224^2$ with a short multi-resolution fine-tuning step to adapt the tokenizers up to resolution $448^2$.

The base resolution training specifics of our tokenizers are shown in Table 3. We employ several common practices: Following Yu et al. [121], we heavily reduce the dimensionality of the codebook and $l_2$-normalize the codes and encoded vectors to improve codebook utilization and reconstruction quality. In addition, to avoid the joint VQ-VAE and diffusion training to collapse to degenerate solution, we found it to be crucial to restart stale codebook entries, similar to Zeghidour et al. [127]. To that end, we count the number of encoded vectors in a batch that map to a given codebook entry after every iteration, and replace (randomly from the batch) any codes that have an exponential moving average (EMA) count less than a specified threshold $\text{thresh}_{\text{replace}}$. Because this value depends on the total batch size $B$, number of tokens per image $N_{\text{tokens}}$, and codebook vocabulary size $N_{\text{vocab}}$, we specify this EMA threshold using a coefficient $c_{\text{replace}}$ by:

$$\text{thresh}_{\text{replace}} = \frac{B N_{\text{tokens}}}{c_{\text{replace}} N_{\text{vocab}}}$$

Table 3: **Tokenizer training settings.** Base resolution ($224^2$) training configuration for the tokenizers used for training 4M. The base learning rate is specified for batch size 256.

| Configuration | RGB | Normals | Depth | Segmentation | CLIP |
|---|---|---|---|---|---|
| Codebook size | 16384 | 8192 | 8192 | 4096 | 8192 |
| Patch size | | | $16 \times 16$ | | |
| Latent dimension | | | 32 | | |
| EMA stale code coef. $c_{\text{replace}}$ | | | 32 | | |
| Codebook EMA | | | 0.99 | | |
| $l_2$-normalized codes [121] | | | ✓ | | |
| Codebook weight | | | 1.0 | | |
| Commitment weight | | | 1.0 | | |
| Encoder architecture | | | ViT-B/16 | | |
| Decoder architecture | Patch-UNet | Patch-UNet | Patch-UNet | ViT-B/16 | ViT-B/16 |
| Diffusion decoder | ✓ | ✓ | ✓ | ✗ | ✗ |
| Loss function | MSE | MSE | Smooth L1 | Cross-entropy | Smooth L1 |
| Optimizer | | | AdamW [73] | | |
| Opt. momentum | | | $\beta_1, \beta_2 = 0.9, 0.95$ | | |
| Weight decay | | | 0.05 | | |
| Base learning rate [46] | | | 5e-5 | | |
| Batch size | 640 | 640 | 640 | 640 | 1024 |
| Learning rate sched. | | | Cosine decay | | |
| Training epochs | | | 100 (IN1K) + 15 (CC12M) | | |
| Warmup epochs | | | 5 (IN1K) + 1 (CC12M) | | |
| Resolution | | | $224^2$ | | |
| Random crop scale | (0.8, 1.0) | (0.8, 1.0) | (0.8, 1.0) | (0.8, 1.0) | (0.2, 1.0) |
| Random crop ratio | | | (0.75, 1.3333) | | |
| Horizontal flip | | | ✓ | | |
| Data type | float16 | float16 | float16 | bfloat16 | bfloat16 |

**Semantic segmentation & CLIP feature maps.** We tokenize segmentation maps and CLIP feature maps using ViT-B [31] encoders and decoders (without the patch projection on the latter). The reconstruction loss for segmentation maps is cross-entropy, while for CLIP we choose the smooth L1 loss. While we do not use any data augmentations for training 4M, we decided to train the CLIP tokenizer using a minimum crop scale of 0.2 to keep open the option of performing fine-tuning with CLIP targets and strong augmentations [82]. For all other tokenizers, we use a more moderate minimum crop scale of 0.8, to avoid training them on very low-resolution but upscaled images that could affect image fidelity negatively.

**RGB, normals & depth.** Training a VQ-VAE by simply minimizing a reconstruction loss can lead to blurry and unrealistic looking results [88], which is why VQ-GAN [35] proposes to use a discriminator to get more realistic results. We found VQ-GAN training to be unstable and decided to instead train a diffusion model as the decoder, similar to DiVAE [101]. Unlike DiVAE, which trains a diffusion model decoder using a frozen and pre-trained VQ-GAN encoder, we train both the encoder and diffusion decoder end-to-end and from scratch. As the decoder, we use a UNet [80] with four down and up layers, each consisting of three residual blocks and one up/downsampling block, with self-attention blocks at the two smallest down and up layers.

For improved training and inference efficiency, we process $C$-channel image patches of size $C \times 4 \times 4$, rather than individual pixels, similar to Patched Diffusion Models [75]. We condition the diffusion decoder by concatenating the 32-dimensional codebook entries of the $14 \times 14$ encoded tokens with the noised input after patching, i.e. we concatenate the noised and patched image of shape $16C \times 56 \times 56$ with the upsampled latent tensor of shape $32 \times 56 \times 56$. The output of the UNet gets reshaped back to $C \times 224 \times 224$ before computing the loss.

We found that predicting the noise leads to undesirable color shifts during inference, which is why we predict the clean image instead. In addition, we found that to avoid the training to collapse to a degenerate solution, it is crucial to restart dead / unused codebook entries [127]. We train the diffusion decoder using 1000 DDPM [51] steps and a linear noise schedule. At inference time, we sample from the decoder using 25 DDIM [105] steps.

**Multi-resolution training.** To train super-resolution specializations of 4M (see Appendix A.1), we need to be able to use the tokenizers both at the base resolution of $224 \times 224$, and at our chosen higher resolution of $448 \times 448$. Unlike convolutional VQ-VAEs, however, our ViT based tokenizers do not perform well on resolutions different from the training resolution. For that reason, we perform multi-resolution training as a short fine-tuning step, and initialize from the weights trained at the base resolution. During training, the resolution of every batch is sampled randomly between 224 and 448, in increments of 32. All tokenizers were fine-tuned for one CC12M epoch, except for RGB, which was fine-tuned for five epochs. We lowered the batch size per GPU to 16, except for semantic segmentation, for which we lowered it to 10. The remaining settings are the same as in Table 3.

## C  Method & Training Details

### C.1  Additional 4M architecture details

For both the ablations and the longer training runs we train 4M models of different sizes. Table 4 lists their respective encoder and decoder depths, model dimension, number of attention heads, as well as the number of trainable parameters (excluding the embedding layers).

**Embedding details.** Modality embeddings are shared between the encoder and decoder. Additionally, as is commonly done for autoregressive models, the parameters of the input and output embeddings (i.e., the final linear layer) of the decoder are shared for sequence modalities. Similar to MAE [48] and MultiMAE [5], the positional and modality embeddings are re-injected into the encoder output before it is passed on to the decoder.

**RGB pixels as input.** All 4M models are trained to accept both RGB pixels and tokenized RGB as input. This is implemented by treating RGB pixels and tokenized RGB as two entirely separate modalities, with RGB pixels being an input-only modality (as it isn't tokenized). This simple approach makes it easy to incorporate additional non-tokenized modalities as input in the future.

**Span masking for sequences.** We perform span masking [86] as follows: Given the probability of masking a token ($p_{\text{mask}}$), we randomly mask out tokens in the sequence and replace each consecutive span of masked-out tokens by a sentinel token (e.g., [S_1], [S_2], [S_3],...). The target sequence then consists of the masked-out spans delimited by the sentinel tokens, followed by a final sentinel token to signal the end of the sequence.

Unlike dense modalities, it is not possible to strictly respect the token budget when sampling from sequences due their variable length. Instead, we treat the token budget as a strict upper bound and mask sequences as follows: For the input, we sample a masking probability $p_{\text{mask}}$ from a uniform distribution and use it for span masking. If the sequence length after masking is greater than the input budget, we progressively increase $p_{\text{mask}}$ until it fits within the assigned budget. For the target, if the sequence does not fit within the budget, we randomly truncate it while also ensuring that the first token of the truncated sequence is a sentinel token.

### C.2  Training details

The training details for the 4M models used for the transfer experiments (Section 3) and the generation results (Section 4) are shown in Table 5 . These hyperparameters were chosen using insights from the ablation study (Section 5), more detailed ablation results are shown in Appendix E. All 4M models were trained using bfloat16 mixed-precision [13]. Training durations varied with model size; 4M-B was trained in 1.5 days on 64 A100 GPUs, 4M-L was trained in 3 days on 128 A100 GPUs, while training 4M-XL took 8 days on 128 A100 GPUs. For 4M-XL, we reduced GPU memory

Table 4: **Model size variants:** The model sizes and naming scheme follow T5 [86]. We exclude the size of the embedding layers in the parameter count.

| Model | Encoder Blocks | Decoder Blocks | Model Dim | Num Heads | Total Params |
|-------|----------------|----------------|-----------|-----------|--------------|
| 4M-Ti | 6 | 6 | 384 | 6 | 24M |
| 4M-S | 8 | 8 | 512 | 8 | 59M |
| 4M-B | 12 | 12 | 768 | 12 | 198M |
| 4M-L | 24 | 24 | 1024 | 16 | 705M |
| 4M-XL | 24 | 24 | 2048 | 32 | 2818M |

consumption through the use of activation checkpointing along with optimizer state and gradient sharding (ZeRO-2 [87]), via PyTorch's Fully Sharded Data Parallel (FSDP).

Table 5: **Pre-training settings.** Training configuration for the 4M models used in the transfer experiments and generation results.

| Configuration | 4M-B | 4M-L | 4M-XL |
|---------------|------|------|-------|
| Training length ($n$ tokens) | | 500B | |
| Warmup length ($n$ tokens) | | 10B | |
| Optimizer | | AdamW [73] | |
| Opt. momentum | | $\beta_1, \beta_2 = 0.9, 0.95$ | |
| Base learning rate [46] | 1e-4 | 1e-4 | 2e-5 |
| Batch size | | 8192 | |
| Weight decay | | 0.05 | |
| Gradient clipping | ✗ | ✗ | 3.0 |
| Learning rate schedule | | Cosine decay | |
| Feedforward activation (Appendix E.6) | | SwiGLU [100] | |
| Input token budget (Appendix E.4) | | 128 | |
| Target token budget (Appendix E.4) | | 128 | |
| Input and target $\alpha$ (Appendix E.4) | | 0.5, 0.5 | |
| Masking strategy (Appendix E.4) | | RGB → All & All → All | |
| Image resolution | | $224^2$ | |
| Augmentation | | None (Center Crop) | |
| Repeated sampling [38] (Appendix E.7) | | 4 | |
| Data type | | bfloat16 [13] | |

# D    Transfer Experiments Details

In this section, we report the fine-tuning details for all transfers shown in Section 3. Whenever possible, we follow commonly used settings from other papers performing such transfers. However, to avoid excessive computational costs, we adjust some transfer settings such that no single transfer requires more than 20% of the pre-training compute. These adjustments include reducing the training resolution when fine-tuning on COCO, and reducing the number of epochs for intermediate fine-tuning on ImageNet-21K.

## D.1    Architectural differences with the baselines

4M 's encoder and all baselines can be viewed as ViT-Base or ViT-Large backbones [31], and are nearly identical in terms of parameter count and model FLOPS. Despite this, a few minor architectural differences could potentially affect their performance. These include: (1) 4M lacks bias terms and uses SwiGLU [100] in the feedforward, while the baselines include bias terms and use GELU [49]. (2) All baselines include an additional learnable [CLS] token as input, which is absent in 4M. (3) DeiT III uses learnable positional embeddings, whereas all other methods including 4M use fixed sine-cosine positional embeddings. (4) When fine-tuning on COCO, MAE uses both absolute and relative positional embeddings (following ViTDet's approach [66]), while all other methods only use absolute positional embeddings.

## D.2 Image classification on ImageNet-1K

We use settings inspired by [114] and first perform intermediate fine-tuning on ImageNet-21K [93], followed by full fine-tuning on ImageNet-1K [29]. Intermediate fine-tuning helps improve overall performance for models that were not originally trained on ImageNet-21K (all 4M models and baselines aside from DeiT III). Details are shown in Table 6.

Table 6: **Image classification settings.** Configuration for intermediate fine-tuning on ImageNet-21K and fine-tuning on ImageNet-1K.

| Configuration | ImageNet-21K | | ImageNet-1K | |
|---|---|---|---|---|
| | Base | Large | Base | Large |
| Fine-tuning epochs | 20 | | 50 | 20 |
| Warmup epochs | 2 | | 2 | |
| Optimizer | AdamW [73] | | AdamW [73] | |
| Opt. momentum | $\beta_1, \beta_2 = 0.9, 0.95$ | | $\beta_1, \beta_2 = 0.9, 0.999$ | |
| Base learning rate [46] | 1e-4 | | 1e-4 | |
| Batch size | 4096 | | 4096 | |
| Weight decay | 0.05 | | 0.05 | |
| Learning rate schedule | Cosine decay | | Cosine decay | |
| Layer-wise lr decay [26] | 0.75 | 0.85 | 0.75 | 0.85 |
| Drop path [55] | 0.1 | 0.2 | 0.1 | 0.2 |
| Input resolution | $224^2$ | | $224^2$ | |
| Augmentation | RandAug(9, 0.5) [27] | | RandAug(9, 0.5) [27] | |
| Random resized crop | (0.5, 1) | | (0.08, 1) | |
| Label smoothing $\varepsilon$ | 0.1 | | 0.1 | |
| Mixup [129] | 0.1 | | 0.1 | |
| Cutmix [124] | 1.0 | | 1.0 | |

## D.3 Object detection and instance segmentation on COCO

We follow the settings from ViTDet [66] with a simplified Cascade Mask-RCNN head [47, 15] with two major changes: First, we reduce the image resolution from 1024×1024 to 512×512, which reduces computational costs during training by over 4× but lowers the performance. Second, we do not use window attention, and instead keep the attention layers unchanged (i.e. global) for all models and rely on activation checkpointing to reduce the GPU memory usage. Details are shown in Table 7.

Table 7: **Object detection and instance segmentation settings.** Configuration for object detection and instance segmentation fine-tuning on COCO, the settings follow ViTDet [66].

| Configuration | COCO | |
|---|---|---|
| | Base | Large |
| Fine-tuning epochs | 100 | |
| Optimizer | AdamW [73] | |
| Opt. momentum | $\beta_1, \beta_2 = 0.9, 0.999$ | |
| Weight decay | 0.1 | |
| Learning rate | 0.0001 | |
| Learning rate schedule | Multi-step decay | |
| Lr schedule milestones | [Epoch 89, Epoch 96] | |
| Lr schedule decay values | [1.0, 0.1, 0.01] | |
| Warmup epochs | 0.01 | 0.007 |
| Batch size | 512 | 320 |
| Layer-wise lr decay [26] | 0.7 | 0.8 |
| Drop path [55] | 0.1 | 0.4 |
| Input resolution | $512^2$ | |
| Augmentation | Large-scale jitter (LSJ) [41] | |

### D.4 Semantic segmentation on ADE20K

We follow the settings from MultiMAE [5] and use a ConvNext [72] prediction head of depth 4 on top of the encoder to perform semantic segmentation. Details are shown in Table 8.

Table 8: **Semantic segmentation settings.** Configuration for semantic segmentation fine-tuning on ADE20K, the settings follow MultiMAE [5].

| Configuration | ADE20K | |
| --- | --- | --- |
| | Base | Large |
| Fine-tuning epochs | 64 | |
| Warmup epochs | 1 | |
| Optimizer | AdamW [73] | |
| Opt. momentum | $\beta_1, \beta_2 = 0.9, 0.999$ | |
| Learning rate | 2e-4 | |
| Batch size | 64 | |
| Weight decay | 0.05 | |
| Learning rate schedule | Cosine decay | |
| Layer-wise lr decay [26] | 0.75 | 0.85 |
| Drop path [55] | 0.1 | 0.2 |
| Input resolution | $512^2$ | |
| Augmentation | Large-scale jitter (LSJ) [41] | |
| Color jitter | ✓ | |

### D.5 Depth estimation on NYUv2

We closely follow the settings from MultiMAE [5] but replace the DPT [91] prediction head by a ConvNeXt [72] prediction of depth 2. We find that this ConvNeXt head reaches comparable performance to DPT on the MultiMAE baseline, while also removing the need to extract and rescale features from intermediate layers of the network. Details are shown in Table 9.

Table 9: **Depth estimation settings.** Configuration for depth estimation fine-tuning on NYUv2, the settings follow MultiMAE [5].

| Configuration | NYUv2 | |
| --- | --- | --- |
| | Base | Large |
| Fine-tuning epochs | 1000 | |
| Warmup epochs | 100 | |
| Optimizer | AdamW [73] | |
| Opt. momentum | $\beta_1, \beta_2 = 0.9, 0.999$ | |
| Learning rate | 1e-4 | |
| Batch size | 128 | |
| Weight decay | 1e-4 | |
| Learning rate schedule | Cosine decay | |
| Layer-wise lr decay [26] | 0.75 | 0.85 |
| Drop path [55] | 0.1 | 0.2 |
| Input resolution | $256^2$ | |
| Random crop | ✓ | |
| Color jitter | ✓ | |

# E Ablation Details & Results

## E.1 Benchmark tasks

All transfer tasks in the ablation are cast as token-to-token prediction problems, similar to the way 4M was pre-trained. We perform several transfers with RGB (pixels) as the input, test how well 4M variants perform at adapting to unseen input modalities, and ablate their ability to make use of multiple input modalities.

All models are fine-tuned with a constant learning rate and evaluated using an exponential moving average of the weights. As different models may overfit to the transfer tasks at different speeds, we measure the validation loss 50 times per transfer and report the best loss, as in T5 [86].

For all tasks and modalities, we train new tokenizers on the respective training sets, which means that they are all either completely unseen, or a new instantiation of a training task. When transferring from an unseen modality (e.g., in X→Y and X+Y→Z transfers), the input embeddings are not initialized, which can lead to instabilities and lower the overall performance. To address this problem, we initially freeze the Transformer encoder-decoder and only train the new input and output embedding layers. Afterwards, we unfreeze the Transformer and train the entire model.

As described in Section 5, we report the validation set cross-entropy performance instead of task-specific metrics. The reasons for this are twofold: First, downstream performance hinges on A) how well the tokens are able to represent the downstream tasks (i.e. tokenizer reconstruction error), and B) how well 4M is able to predict these tokens (i.e. cross-entropy loss). Since the tokenizers are the same across all settings, we abstract away aspect A for this ablation and only report B. Second, reporting the cross-entropy loss makes comparisons more uniform and avoids having to scale and average wildly different task-specific metrics such as mAP, mIoU, MSE, etc. that are not comparable with each other.

**COCO transfers.** We perform object detection on the COCO dataset [67] in the same manner as during pre-training (see Appendix B.2), i.e. by casting it as a sequence prediction problem [21]. The input modality for the transfer task is RGB (pixels). Exact training settings are provided in Table 10.

**ADE20K transfers.** We perform semantic segmentation on ADE20K [131] by fine-tuning the 4M models to predict VQ-VAE tokens of the targets. Since our semantic segmentation tokenizer during pre-training was trained with COCO classes, we train a new tokenizer for ADE20K in the same manner (see Appendix B.2). The tokenizer is trained for 1500 epochs with a warmup period of 150 epochs. In order to perform random crop augmentations at transfer time, we also pre-trained the tokenizer with a crop scale of (0.2, 1.0). The input modality for the transfer task is RGB (pixels). Exact training settings are provided in Table 11.

Table 10: **COCO detection transfer settings.**

| Configuration | COCO Det. |
|---|---|
| Fine-tuning epochs | 100 |
| Warmup epochs | 5 |
| Optimizer | AdamW [73] |
| Opt. momentum | $\beta_1, \beta_2 = 0.9, 0.999$ |
| Base learning rate [46] | 1e-4 |
| Batch size | 256 |
| Weight decay | 0.05 |
| Learning rate schedule | Constant |
| EMA decay | 0.998 |
| Eval. freq (epochs) | 2 |
| Input resolution | $224^2$ |
| Augmentation | Large-scale jitter (LSJ) [41] |
| Color jitter | ✓ |

Table 11: **ADE20K semantic segmentation transfer settings.**

| Configuration | ADE20K Seg. |
|---|---|
| Fine-tuning epochs | 200 |
| Warmup epochs | 20 |
| Optimizer | AdamW [73] |
| Opt. momentum | $\beta_1, \beta_2 = 0.9, 0.999$ |
| Base learning rate [46] | 1e-4 |
| Batch size | 256 |
| Weight decay | 0.5 |
| Learning rate schedule | Constant |
| EMA decay | 0.998 |
| Eval. freq (epochs) | 4 |
| Input resolution | $224^2$ |
| Augmentation | Random crop |
| Crop scale | (0.2, 1.0) |
| Crop ratio | (0.75, 1.333) |
| Color jitter | ✗ |

**Taskonomy-20K & Hypersim RGB→X transfers.** We perform several transfers on Hypersim [94] and a subset of Taskonomy [126] Tiny, denoted here as Taskonomy-20K. The subset contains 20'000 training images and 2000 validation images.

For Taskonomy-20K, we transfer to `depth`, `principal curvature`, `reshading`, `occlusion edges`, `2D edges`, `2D keypoints`, and `3D keypoints`. For Hypersim, we transfer to `surface normals`, `semantic segmentation`, and `2D edges`. The input modality for all transfer tasks is RGB (pixels). Exact training settings are provided in Table 12.

**Taskonomy-20K & Hypersim X→Y transfers.** There are 72 possible transfers between the Taskonomy tasks, and 12 possible transfers between the Hypersim tasks. To reduce the computational cost of the ablations, we subsample these transfer tasks, trying to cast a diverse net. For Taskonomy-20K, we perform `curvature → 2D keypoints`, `depth → normals`, `2D edges → depth`, `surface normals → 2D edges`, `occlusion edges → principal curvature`, and `RGB (tokens) → surface normals`. For Hypersim we perform `RGB (tokens) → segmentation`, `2D edges → segmentation`, `normals → segmentation`, `segmentation → 2D edges`, and `segmentation → normals`. Exact training settings are provided in Table 13.

**Taskonomy-20K & Hypersim X+Y→Z transfers.** As before, we subsample all possible transfer tasks that use two modalities as the input, aiming for a diverse set of transfers. For Taskonomy-20K, we perform `RGB (pixels) + normals → depth`, `RGB (pixels) + depth → reshading`, `reshading + 2D keypoints → normals`, `2D edges + 3D keypoints → RGB (tokens)`, `2D edges + curvature → occlusion edges`, and `occlusion edges + 2D keypoints → curvature`. For Hypersim we perform `RGB (pixels) + depth → segmentation`, `RGB (pixels) + normals → segmentation`, `RGB (pixels) + segmentation → depth`, `depth + segmentation → RGB (tokens)`, `2D keypoints + normals → segmentation`, and `2D edges + depth → segmentation`. Exact training settings are provided in Table 13.

Table 12: **RGB→X transfer settings.**

| Configuration | Taskonomy-20K | Hypersim |
|---|---|---|
| Fine-tuning epochs | 100 | 50 |
| Warmup epochs | 5 | 2 |
| Frozen Transformer epochs | 0 | 2 |
| Optimizer | AdamW [73] | |
| Opt. momentum | $\beta_1, \beta_2 = 0.9, 0.999$ | |
| Base learning rate [46] | 1e-4 | |
| Frozen Transformer base lr | 1e-4 | |
| Batch size | 256 | |
| Weight decay | 0.05 | |
| Learning rate schedule | Constant | |
| EMA decay | 0.998 | |
| Eval. freq (epochs) | 2 | 1 |
| Input resolution | $224^2$ | |
| Augmentation | Random crop | |
| Crop scale | (0.2, 1.0) | |
| Crop ratio | (0.75, 1.333) | |
| Color jitter | ✗ | ✓ |

Table 13: **X→Y and X+Y→Z transfer settings.**

| Configuration | Taskonomy-20K | Hypersim |
|---|---|---|
| Fine-tuning epochs | 100 | 50 |
| Warmup epochs | 5 | 2 |
| Frozen Transformer epochs | 25 | 20 |
| Optimizer | AdamW [73] | |
| Opt. momentum | $\beta_1, \beta_2 = 0.9, 0.999$ | |
| Base learning rate [46] | 1e-4 | |
| Frozen Transformer base lr | 2e-3 | |
| Batch size | 256 | |
| Weight decay | 0.05 | |
| Learning rate schedule | Constant | |
| EMA decay | 0.998 | |
| Eval. freq (epochs) | 2 | 1 |
| Input resolution | $224^2$ | |
| Augmentation | Random crop | |
| Crop scale | (0.2, 1.0) | |
| Crop ratio | (0.75, 1.333) | |
| Color jitter | Only if RGB is in the input | |

### E.2 Reference model

In this section, we specify the reference model and its training settings. These choices are made by following common practices and performing educated guesses. The settings include model size, number of encoding and decoding layers, additional modifications on the architecture (e.g. no bias), choice of Dirichlet sampling parameter $\alpha$, input and target modalities, and training length. Following common practice for the base model size [30, 86], our base model consists of 12 encoder and 12 decoder layers. We train it on CC12M [19] using all modalities as both inputs and targets, at resolution $224 \times 224$ pixels (corresponding to $14 \times 14$ tokens per dense modality), and using *no augmentations* such as cropping nor color augmentations. The total training length is fixed at 100B tokens seen, corresponding to roughly 400M masked samples seen for the base setting. We set the number of randomly sampled input and target tokens both to 128, and sample each using a symmetric Dirichlet distribution with parameter $\alpha = 0.2$. Full details are shown in Table 14.

To determine the significance of various modeling choices, we adopt the approach used by Raffel et al. [86] that calculates the standard deviation of the transfer results of ten distinct reference models, each trained with different seeds. In the subsequent ablation results, the average performance of the reference model setting is indicated by the symbol ▷. Furthermore, we highlight transfer results falling within two standard deviations (shown in Table 15) of the lowest achieved losses, enabling us to estimate which settings are significant. Table 15 further contains from-scratch results (training a randomly initialized model) for all the transfer tasks. This serves as a lower bound for the performance

that can be achieved on a given transfer task, and allows us to calibrate how difficult each transfer is, and by how much pre-training with a given setting improves upon random initialization.

Table 14: **Reference model pre-training settings.** Training configuration for the reference 4M model used in the ablation. All other models in ablation are obtained by changing just one of these hyperparameters.

| Configuration | Reference model |
|---|---|
| Model size | Base (see Table 4) |
| Training length ($n$ tokens) | 100B |
| Warmup length ($n$ tokens) | 10B |
| Optimizer | AdamW [73] |
| Opt. momentum | $\beta_1, \beta_2 = 0.9, 0.95$ |
| Base learning rate [46] | 1e-4 |
| Batch size | 4096 |
| Weight decay | 0.05 |
| Gradient clipping | ✗ |
| Learning rate schedule | Cosine decay |
| Feedforward activation | GELU [49] |
| Input token budget | 128 |
| Target token budget | 128 |
| Input and target $\alpha$ | 0.2, 0.2 |
| Masking strategy | All $\rightarrow$ All |
| Image resolution | $224^2$ |
| Augmentation | None (Center Crop) |
| Repeated sampling [38] | 4 |

Table 15: **Reference model results:** We report the average and standard deviation of the transfer results achieved by our reference 4M model. In addition, we show performance achieved by a randomly initialized model. All results are reported on the validation sets of each dataset.

| | COCO Det. | ADE20K Seg. | RGB$\rightarrow$X | X$\rightarrow$Y | X+Y$\rightarrow$Z | Avg. Loss |
|---|---|---|---|---|---|---|
| ▷ Reference model avg. | **3.06** | **6.11** | **5.07** | **6.75** | **5.80** | **5.36** |
| No pre-training | 3.31 | 6.59 | 6.03 | 7.52 | 6.62 | 6.01 |
| Reference model std. dev. | 0.008 | 0.008 | 0.021 | 0.018 | 0.016 | 0.010 |

### E.3 Input modalities and target tasks

**Importance of target tasks for representation learning.** 4M is both a multimodal and a *multitask* pre-training scheme and the choice of target task(s) is a powerful way of steering what representation the model learns [10, 109]. Furthermore, multitask pre-training has been shown to improve downstream task transfers [42, 107, 5]. In this ablation, we would like to learn how the choice of target tasks affects transfers to various downstream tasks. For that, we fix the input modalities to be either RGB or all the modalities (denoted as *"All"*), and vary just the target tasks. The results in Table 16 (or equivalently, Table 2) mirror the findings of Sax et al. [98], that the optimal choice of pre-training setting depends highly on the type of transfer that is performed, and that there is no single pre-training task that performs best on all transfers. That said, pre-training on all target modalities outperforms the other single-task and multitask alternatives in terms of average loss, no matter what input modalities were used during pre-training. Note that some of the settings in Table 16 are conceptually similar to well-known pre-training methods, and can serve as baselines for such methods, while abstracting away implementation details that would make them difficult to compare otherwise. E.g., RBG $\rightarrow$ RGB (first row) is similar to MAE [48], while RGB $\rightarrow$ CLIP (fifth row) is similar to BEITv2 [82].

**Importance of multimodal pre-training for transferring to new modalities.** Just as multitask pre-training improves transfers to new target tasks, we would hope that, likewise, multimodal pre-training improves transfers to new input modalities. Indeed, pre-training with multiple modalities

that may be available at transfer time has been shown to be beneficial [5], but it is not as clear how well multimodal models transfer to completely new, unseen input modalities. Table 16 shows that multimodal pre-training can significantly help with transferring to new input modalities (X→Y transfers), but comes at a performance loss at transfers that use RGB as the sole input modality. In Appendix E.4 we will explore pre-training using mixtures of different masking strategies, which enables us to train models that perform well in both regimes.

Table 16: **Pre-training input and target modalities ablation:** The choice of pre-training tasks and modalities influences what representations the model learns, and how well it can be transferred to novel tasks and modalities. The average rank and best losses are shown for *"RGB"* and *"All"* inputs separately. Here, *Geometric = RGB + Depth + Normals* and *Semantic = RGB + Segmentation + CLIP + Detection + Captions*. Performing 4M pre-training on all input and target modalities is the most versatile choice, if the optimal set of pre-training modalities for any given downstream task is unknown. Note that the results shown here are identical to those of Table 2, and are included for completeness.

| Pre-training inputs | Pre-training targets | COCO Det. | ADE20K Seg. | RGB→X | X→Y | X+Y→Z | Avg. Loss | Avg. Rank |
|---|---|---|---|---|---|---|---|---|
| RGB | RGB | 3.14 | 6.21 | 5.03 | 6.94 | 6.15 | 5.49 | 8.00 |
| RGB | Depth | 3.11 | 6.06 | 4.72 | 6.89 | **5.84** | 5.32 | 4.00 |
| RGB | Normals | 3.12 | 6.02 | **4.66** | **6.83** | 5.87 | **5.30** | 3.20 |
| RGB | Segmentation | 3.17 | **5.94** | 4.84 | **6.86** | 5.89 | 5.34 | 4.60 |
| RGB | CLIP | 3.07 | 6.11 | 4.83 | **6.85** | 5.94 | 5.36 | 4.80 |
| RGB | Detection | **2.78** | 6.11 | 5.03 | 7.07 | 6.24 | 5.45 | 7.20 |
| RGB | Captions | 3.45 | 6.55 | 5.92 | 7.35 | 6.86 | 6.03 | 10.00 |
| RGB | Geometric | 3.11 | 6.08 | **4.70** | 6.88 | **5.85** | 5.32 | 3.80 |
| RGB | Semantic | 2.88 | 5.99 | 4.86 | 6.97 | 6.06 | 5.35 | 5.40 |
| RGB | All | 2.90 | 5.99 | 4.74 | 6.91 | 5.93 | **5.29** | 4.00 |
| All | RGB | 3.21 | 6.20 | 5.07 | **6.75** | 5.85 | 5.42 | 3.80 |
| All | CLIP | 3.19 | 6.18 | 5.06 | 6.80 | 5.88 | 5.42 | 3.80 |
| All | Geometric | 3.20 | **6.13** | **4.98** | **6.72** | **5.76** | **5.36** | 1.80 |
| All | Semantic | **3.05** | 6.13 | 5.16 | 6.77 | 5.87 | 5.39 | 3.40 |
| ▷ All | All | **3.06** | **6.11** | 5.07 | **6.75** | 5.80 | **5.36** | 2.20 |

## E.4 Multimodal masking strategy

Multimodal masking is at the core of 4M, so in this section, we ablate how modality tokens should be sampled and mixed, and how many tokens we should encode and decode. When performing multimodal training, it can be computationally challenging to deal with the large number of tokens from all modalities. For that reason, the choice of the number of input and target tokens has the potential to greatly reduce the computational burden, and setting them as low as possible is desirable.

**Mask sampling parameter.** The number of tokens to sample from each modality is determined by a symmetric Dirichlet distribution with parameter $\alpha$ [5]. If $\alpha$ is low, the sampling procedure will often choose cases where most of the tokens are sampled from only one modality. If $\alpha$ is high, however, most samples will contain tokens from all modalities to equal proportions. We ablate here various choices of $\alpha$ values, both for the input and target sampling. Table 17 shows that models trained with higher $\alpha$ values generally transfer better to RGB tasks, but methods trained with lower input $\alpha$ values perform better at transferring to novel input modalities.

Table 17: **Mask sampling parameter ablation:** The choice of Dirichlet sampling parameter $\alpha$ influences how many tokens are sampled from each modality. Low values lead to many samples consisting of entire modalities, while high values result in equal numbers of tokens to be sampled from each modality. We ablate various input and target $\alpha$ values.

| Input $\alpha$ | Target $\alpha$ | COCO Det. | ADE20K Seg. | RGB→X | X→Y | X+Y→Z | Avg. Loss | Avg. Rank |
|---|---|---|---|---|---|---|---|---|
| 0.1 | 0.1 | 3.11 | 6.14 | 5.18 | 6.81 | 5.89 | 5.43 | 4.80 |
| ▷ 0.2 | 0.2 | 3.06 | 6.11 | 5.07 | **6.75** | **5.80** | 5.36 | 3.20 |
| 0.5 | 0.5 | 3.02 | 6.09 | 4.98 | **6.76** | **5.80** | **5.33** | 2.60 |
| 1.0 | 1.0 | **3.00** | 6.07 | **4.95** | 6.80 | **5.79** | **5.32** | 1.80 |
| ∞ | ∞ | **3.00** | 6.08 | **4.93** | 6.84 | 5.83 | **5.34** | 2.60 |

**Input masking budget.** The difficulty of the multimodal masked modeling task is largely determined by the number of visible (non-masked) input tokens, with fewer tokens used making the task more challenging. Indeed, since the modalities contain a lot of spatial information about each other, it is necessary to lower the number of visible tokens to keep the objective difficult enough. Furthermore, encoding only a small set of visible tokens (as opposed to encoding both visible *and* mask tokens [30, 7]) can significantly improve training efficiency [48]. This is especially important when training on multiple modalities [5], as the input length would otherwise grow too large.

Table 18: **Input masking budget ablation:** The number of visible input tokens decides the difficulty of the masked modeling task (the fewer given, the harder the task), and influences the computational cost of the encoder (the fewer, the less expensive). We fix the target masking budget at 128 tokens and vary the number of input tokens.

| Input Tokens | COCO Det. | ADE20K Seg. | RGB→X | X→Y | X+Y→Z | Avg. Loss | Avg. Rank |
|---|---|---|---|---|---|---|---|
| 64 | 3.15 | 6.21 | 5.27 | 6.89 | 5.99 | 5.50 | 4.00 |
| ▷ 128 | 3.06 | 6.11 | **5.07** | 6.75 | 5.80 | 5.36 | 2.00 |
| 256 | **3.03** | **6.08** | **5.04** | **6.71** | **5.75** | **5.32** | 1.00 |
| 512 | 3.07 | 6.12 | 5.11 | 6.78 | 5.83 | 5.38 | 3.00 |

**Target masking budget.** Just like how we could improve the training efficiency by only decoding a small set of visible tokens, so can we decide to decode merely a subset of all remaining masked-out tokens. Decoding all masked-out tokens, like MAE [48], or in the multimodal case MultiMAE [5], can quickly become infeasible as the number of modalities grows and the masking ratio is kept high. While they attempt to address this issue by decreasing the size of the decoder, such a strategy might ultimately hurt generation quality if the decoder is used for generative tasks. As Table 19 shows, decoding only a small random subset of all targets performs better (for a fixed training duration) than decoding a larger number of tokens, while also significantly reducing the computational costs.

Table 19: **Target masking budget ablation:** Similar to passing only visible tokens to the encoder, we can decide to only decode a subset of masked-out tokens, which improves computational efficiency considerably. We fix the input masking budget at 128 and vary the number of target tokens.

| Target Tokens | COCO Det. | ADE20K Seg. | RGB→X | X→Y | X+Y→Z | Avg. Loss | Avg. Rank |
|---|---|---|---|---|---|---|---|
| 64 | **3.06** | **6.11** | **5.09** | **6.73** | **5.79** | **5.36** | 1.40 |
| ▷ 128 | **3.06** | **6.11** | **5.07** | 6.75 | 5.80 | **5.36** | 1.60 |
| 256 | 3.10 | 6.16 | 5.15 | 6.81 | 5.91 | 5.43 | 3.00 |
| 512 | 3.16 | 6.20 | 5.27 | 6.84 | 5.95 | 5.48 | 4.00 |

**Mixture of masking strategies.** As shown in Appendix E.3, pre-training using RGB as the sole input modality instead of training on all of them performs significantly better at transfers that, likewise, use RGB as input. On the flip-side, pre-training with all modalities performs significantly better when transferring to unseen input modalities. This difference can be explained by the fact that these two mask sampling strategies are on opposite extremes – the number of RGB inputs in the multimodal pre-training case is relatively small, while the RGB-only case is never trained on any other modalities. We can find a compromise by sampling batch elements using these two strategies uniformly at random, creating a pre-training strategy, where approximately half the time input tokens are RGB-only, and half the time they are sampled from all modalities. Table 20 shows that while the mixture approach does not perform better than either individual mask sampling strategy at any of the transfers, it is a good compromise if the use-case of the model is open at the time of pre-training.

Table 20: **Mixture of masking strategies ablation:** Mixing different masking strategies by randomly sampling between the two provides a good compromise in downstream performance.

| Masking strategy | COCO Det. | ADE20K Seg. | RGB→X | X→Y | X+Y→Z | Avg. Loss | Avg. Rank |
|---|---|---|---|---|---|---|---|
| RGB → All | **2.90** | **5.99** | **4.74** | 6.91 | 5.93 | **5.29** | 1.80 |
| ▷ All → All | 3.06 | 6.11 | 5.07 | **6.75** | **5.80** | 5.36 | 2.20 |
| RGB → All & All → All | 2.95 | **6.00** | 4.86 | 6.86 | 5.87 | **5.31** | 2.00 |

### E.5 How well does 4M scale?

We argue that scalability is a key property that models and training objectives should have. To ablate how well 4M scales, we ablate the following three axes: dataset size, training length, and model size.

**Does 4M benefit from more data?**  We train the base configuration on various subsets of CC12M, down to 1/64th of the full dataset. The results in Table 21 show that 4M is able to scale with the pre-training dataset size.

Table 21: **Dataset size ablation:** To estimate how well 4M scales with pre-training dataset size, we train it on different subsets of CC12M ranging from $\frac{1}{64}$ of the dataset to the entire dataset..

| Dataset fraction | COCO Det. | ADE20K Seg. | RGB→X | X→Y | X+Y→Z | Avg. Loss | Avg. Rank |
|---|---|---|---|---|---|---|---|
| 1/64 | 3.22 | 6.23 | 5.31 | 6.86 | 5.94 | 5.51 | 4.00 |
| 1/16 | 3.15 | 6.17 | 5.23 | 6.80 | 5.87 | 5.44 | 3.00 |
| 1/4 | 3.08 | 6.13 | **5.11** | 6.79 | 5.84 | 5.39 | 2.00 |
| ▷ 1 | **3.06** | **6.11** | **5.07** | **6.75** | **5.80** | **5.36** | 1.00 |

**Does 4M benefit from longer training?**  We train the base configuration for different amounts of total tokens seen. We define *tokens seen* as the total number of both input tokens and target tokens the model was trained on. Table 22 shows that 4M scales to longer training schedules.

Table 22: **Training duration ablation:** To see whether 4M benefits from longer training schedules, we train it for up to 400B tokens seen.

| Train tokens [B] | COCO Det. | ADE20K Seg. | RGB→X | X→Y | X+Y→Z | Avg. Loss | Avg. Rank |
|---|---|---|---|---|---|---|---|
| 50 | 3.13 | 6.17 | 5.23 | 6.86 | 5.93 | 5.46 | 4.00 |
| ▷ 100 | 3.06 | 6.11 | 5.07 | 6.75 | 5.80 | 5.36 | 3.00 |
| 200 | 3.00 | 6.05 | 4.93 | **6.71** | **5.74** | 5.29 | 2.00 |
| 400 | **2.96** | **6.02** | **4.84** | **6.70** | **5.71** | **5.25** | 1.00 |

**Does 4M scale with model size?**  We train 4M models of different sizes, ranging from Tiny (4M-Ti) to Large variants (4M-L). Exact model specifications are given Table 4. Table 23 shows that 4M scales with model size.

Table 23: **Model size ablation:** To see how 4M scales with model size, we train Tiny, Small, Base and Large versions.

| Model size | COCO Det. | ADE20K Seg. | RGB→X | X→Y | X+Y→Z | Avg. Loss | Avg. Rank |
|---|---|---|---|---|---|---|---|
| 4M-Ti | 3.28 | 6.38 | 5.57 | 6.93 | 6.15 | 5.66 | 4.00 |
| 4M-S | 3.16 | 6.25 | 5.34 | 6.84 | 5.97 | 5.51 | 3.00 |
| ▷ 4M-B | 3.06 | 6.11 | 5.07 | 6.75 | 5.80 | 5.36 | 2.00 |
| 4M-L | **2.93** | **5.97** | **4.86** | **6.63** | **5.75** | **5.23** | 1.00 |

### E.6 Architectural design choices

In the following, we ablate several design choices related to the Transformer architecture.

**Number of encoder & decoder layers.** We ablate whether one should train 4M using a balanced allocation of encoder and decoder layers or rather go with a deeper or more shallow decoder. We fix the total number of Transformer layers to 24 and ablate three different configurations in Table 24. The results show that all configurations perform comparably, with the balanced and encoder-heavy settings having a slight edge over the decoder-heavy one.

Table 24: **Encoder & decoder depth ablation:** We ablate three different ways of allocating 24 Transformer layers between the encoder and decoder.

| Enc. depth | Dec. depth | COCO Det. | ADE20K Seg. | RGB→X | X→Y | X+Y→Z | Avg. Loss | Avg. Rank |
|---|---|---|---|---|---|---|---|---|
| 8 | 16 | 3.10 | 6.14 | **5.09** | **6.75** | 5.82 | 5.38 | 2.80 |
| ▷ 12 | 12 | **3.06** | **6.11** | 5.07 | **6.75** | 5.80 | **5.36** | 1.40 |
| 16 | 8 | **3.05** | **6.11** | 5.11 | **6.75** | 5.78 | **5.36** | 1.80 |

**Per-token vs. per-modality loss.** Since we predict multiple modalities at the same time, there are several ways we can compute their losses. We ablate the following two loss weighting strategies: (1) We treat every predicted token the same and average all their losses. We call this setting *per-token loss*. This setting is biased against modalities that don't contain a lot of tokens, such as captions. (2) We first average the loss for every target modality individually and then average those. We call this setting *per-modality loss*. Computing the loss per-modality noticeably outperforms the per-token loss, as shown in Table 25.

Table 25: **Multitask loss aggregation ablation:** We ablate weighting the loss contribution of every modality equally vs. weighting every token equally.

| Loss type | COCO Det. | ADE20K Seg. | RGB→X | X→Y | X+Y→Z | Avg. Loss | Avg. Rank |
|---|---|---|---|---|---|---|---|
| ▷ Per-modality loss | **3.06** | **6.11** | **5.07** | **6.75** | **5.80** | **5.36** | 1.00 |
| Per-token loss | 3.07 | 6.16 | 5.12 | 6.78 | 5.86 | 5.40 | 2.00 |

**Transformer architecture modifications.** We ablate several modifications to the Transformer architecture, and show results in Table 26. (1) Following Shazeer [100], we ablate the use of the SwiGLU activation function in the feed-forward layers, and reduce the feed-forward dimension by a factor of $\frac{2}{3}$ (from $4d$ to $\frac{2}{3}4d$) to keep the parameter count and amount of computation constant. (2) Following T5 [86] and PaLM [25], we ablate removing all bias terms from the Transformer. (3) Following ViT-22B [28], we ablate the use of query/key normalization, which has been shown to help the stability of very large Vision Transformers trained to perform image classification.

We find SwiGLU slightly outperforms GELU while also removing the need for bias terms, and therefore use it to train the final model. As we do not observe noticeable stability issues with our training objective with current model scales, we refrain from using query/key normalization as we find that it negatively impacts the transfer performance.

Table 26: **Architecture modifications ablation:** We ablate the use of different activations in the feed-forward [49, 100], removing bias terms [86, 25], and performing query-key normalization [28].

| FFN act. | Bias | QK Norm. | COCO Det. | ADE20K Seg. | RGB→X | X→Y | X+Y→Z | Avg. Loss | Avg. Rank |
|---|---|---|---|---|---|---|---|---|---|
| ▷ GELU | ✓ | ✗ | **3.06** | 6.11 | **5.07** | **6.75** | **5.80** | **5.36** | 2.20 |
| GELU | ✗ | ✗ | 3.07 | 6.11 | 5.11 | 6.80 | 5.86 | 5.39 | 4.00 |
| GELU | ✗ | ✓ | 3.08 | 6.13 | 5.16 | 6.92 | 5.96 | 5.45 | 5.20 |
| SwiGLU | ✓ | ✗ | 3.08 | 6.13 | 5.16 | 6.94 | 6.26 | 5.51 | 5.80 |
| SwiGLU | ✗ | ✗ | 3.07 | 6.11 | **5.03** | **6.73** | **5.79** | **5.34** | 1.80 |
| SwiGLU | ✗ | ✓ | **3.06** | **6.09** | **5.03** | 6.79 | **5.80** | **5.35** | 2.00 |

### E.7 Training design choices

**Base learning rate.** We ablate several base learning rates [46] and show results in Table 27.

Table 27: **Base learning rate ablation.** We ablate the choice of learning rate used during training. 4M is not too sensitive to the choice of learning rate, although we observe some performance degradation and instabilities when strongly increasing the learning rate.

| Base lr [46] | COCO Det. | ADE20K Seg. | RGB→X | X→Y | X+Y→Z | Avg. Loss | Avg. Rank |
|---|---|---|---|---|---|---|---|
| 5e-5 | **3.06** | **6.11** | **5.07** | 6.78 | **5.82** | **5.37** | 2.80 |
| ▷ 1e-4 | **3.06** | **6.11** | **5.07** | 6.75 | **5.80** | **5.36** | 1.20 |
| 2e-4 | **3.05** | **6.11** | **5.10** | 6.75 | 5.86 | **5.37** | 2.20 |
| 3e-4 | 3.07 | **6.11** | 5.11 | **6.77** | 5.88 | 5.39 | 3.80 |
| 5e-4 | 3.10 | 6.14 | 5.17 | 6.81 | 5.89 | 5.42 | 5.00 |

**Training with repeated sampling.** Data loading can be a significant bottleneck when training efficient masked models. For that reason, we use `webdataset` [116] to load tar files consisting of 1000 samples instead of loading individual samples. We keep a buffer of loaded images in RAM and randomly sample from it repeatedly – each time applying different random masking. Whenever an element in the buffer has been sampled more than the specified number of repeats, we replace it with a fresh sample. Using repeated sampling [38] has the potential to significantly improve training efficiency at no loss in performance (see Table 28).

Table 28: **Repeated sampling ablation:** To improve training efficiency, we reuse samples from a buffer multiple times, at no significant loss in performance.

| Num. repeats | COCO Det. | ADE20K Seg. | RGB→X | X→Y | X+Y→Z | Avg. Loss | Avg. Rank |
|---|---|---|---|---|---|---|---|
| 1 | **3.06** | **6.11** | **5.06** | **6.73** | **5.80** | **5.35** | 1.40 |
| ▷ 4 | **3.06** | **6.11** | **5.07** | 6.75 | **5.80** | **5.36** | 1.60 |
| 8 | 3.07 | 6.12 | 5.10 | 6.76 | 5.81 | 5.37 | 3.00 |

### E.8 Self-baselines

When comparing 4M to other pre-training methods in the transfer learning study, we also report the performance of two self-baselines to better control for the dataset, augmentations, model architecture, compute, and use of tokenizers, all of which can significantly affect downstream performance. These two self-baselines were chosen to be conceptually similar to MAE [48] (Masked RGB → RGB) and BEiT-v2 [82] (Masked RGB → CLIP). Following these methods, we train these self-baselines with a relatively high masking ratio (in our case, 50%) and *ensure that the target tokens do not spatially overlap with the inputs*. In this section, we ablate the masking strategy for these self-baselines by also training a version with a lower masking ratio and with spatial overlap between the input and the targets (as in 4M), and a version without any masking and where all targets are predicted (standard RGB→X training).

Results are shown in Table 29. We find that while the RGB → RGB setting does benefit from higher masking ratios and no spatial overlap (as shown in MAE), RGB → CLIP benefits from a lower masking ratio and from allowing targets to spatially overlap with the input.

Table 29: **Self-baselines ablation:** The RGB → RGB self-baseline performs best with a higher masking ratio and no spatial overlap between inputs and targets, while RGB → CLIP benefits from a lower masking ratio and spatial overlap.

| Target mod. | Input tok. | Target tok. | Spatial overlap | COCO Det. | ADE20K Seg. | RGB→X | X→Y | X+Y→Z | Avg. Loss | Avg. Rank |
|---|---|---|---|---|---|---|---|---|---|---|
| RGB | 98 | 98 | ✗ | **3.12** | **6.16** | **5.01** | 6.91 | **6.14** | **5.47** | 1.40 |
| RGB | 128 | 128 | ✓ | 3.14 | 6.21 | **5.03** | 6.94 | **6.15** | 5.49 | 2.40 |
| RGB | 196 | 196 | ✓ | 3.24 | 6.37 | 5.42 | **6.87** | 6.13 | 5.61 | 2.20 |
| CLIP | 98 | 98 | ✗ | 3.08 | **6.09** | 4.93 | 6.94 | 6.03 | 5.41 | 2.40 |
| CLIP | 128 | 128 | ✓ | **3.07** | 6.11 | **4.83** | **6.85** | **5.94** | **5.36** | 1.20 |
| CLIP | 196 | 196 | ✓ | 3.12 | 6.20 | 4.93 | 6.88 | 6.01 | 5.43 | 2.40 |

### E.9 Comparison to the final models

We gather insights from the ablation to train the "final" 4M models described in Appendix C. Here, we compare these models to the reference models used throughout the ablation. Unsurprisingly, we find that these models perform significantly better than the reference models on the benchmark tasks. The large performance gap for tasks that take as input RGB images can in large part be attributed to the mixture of masking strategies described in Appendix E.4.

Table 30: **Comparison to the final models:** The final models obtained by combining insights from the ablations significantly outperform the reference models on all benchmark tasks.

| Size | Setting | COCO Det. | ADE20K Seg. | RGB→X | X→Y | X+Y→Z | Avg. Loss |
|------|---------|-----------|-------------|-------|-----|-------|-----------|
| ▷ Base | Reference (Table 14) | 3.06 | 6.11 | 5.07 | **6.75** | 5.80 | 5.36 |
| Base | Final (Table 5) | **2.85** | **5.91** | **4.69** | 6.74 | **5.76** | **5.19** |
| Large | Reference (Table 14) | 2.93 | 5.97 | 4.86 | **6.63** | 5.75 | 5.23 |
| Large | Final (Table 5) | **2.68** | **5.76** | **4.48** | 6.67 | **5.63** | **5.04** |

## F  Additional Evaluations

### F.1  Out of the box (zero-shot) performance

In this section, we evaluate the out of the box capabilities of 4M models in three RGB→X tasks: surface normals estimation, depth estimation, and semantic segmentation. The results are shown in Table 31. Our findings demonstrate that 4M performs competitively in these tasks even without fine-tuning, when compared to both task-specific baselines and the pseudo labelers used for 4M training. Additionally, the tokenization quality, as shown in Table 31 and Figure 20, does not seem to limit the overall performance of 4M in these zero-shot scenarios. We therefore anticipate that the zero-shot performance of 4M can be further improved if not limited by the current pseudo labels, either through the use of ground truth data or stronger pseudo labeling networks.

Table 31: **Out of the box (zero-shot) performance:** We report the out of the box performance of 4M models and of baselines on several tasks, all at $224 \times 224$ resolution. We use the DIODE [111] validation set for normals and depth, and COCO validation set for semantic segmentation. **[P]** denotes the pseudo labeler used to train 4M. Best results are **bolded**, second best are underlined. 4M's zero-shot performance matches or outperforms strong baselines such as OASIS [23] and MiDaS [90], and is competitive with pseudo labelers. * To estimate the performance upper bound of 4M due to tokenization, we also report the tokenizer reconstruction quality on validation images from CC12M.

| Method | Surface Normals (mean angle error)↓ | Depth (standardized L1 error)↓ | Semantic Segmentation (mean IoU)↑ |
|--------|------------------------------------|-------------------------------|-----------------------------------|
| OASIS | 34.3 | - | - |
| MiDaS DPT Hybrid | - | 0.73 | - |
| Omnidata+3DCC **[P]** | 22.5 | **0.68** | - |
| Mask2Former Swin-S | - | - | 44.6 |
| Mask2Former Swin-B **[P]** | - | - | 45.7 |
| Mask2Former Swin-L | - | - | **48.0** |
| 4M-B | 21.9 | 0.71 | 41.3 |
| 4M-L | 21.4 | 0.69 | 45.8 |
| 4M-XL | **20.8** | **0.68** | 46.5 |
| Tokenizer reconstruction* | 4.0 | 0.06 | 90.5 |

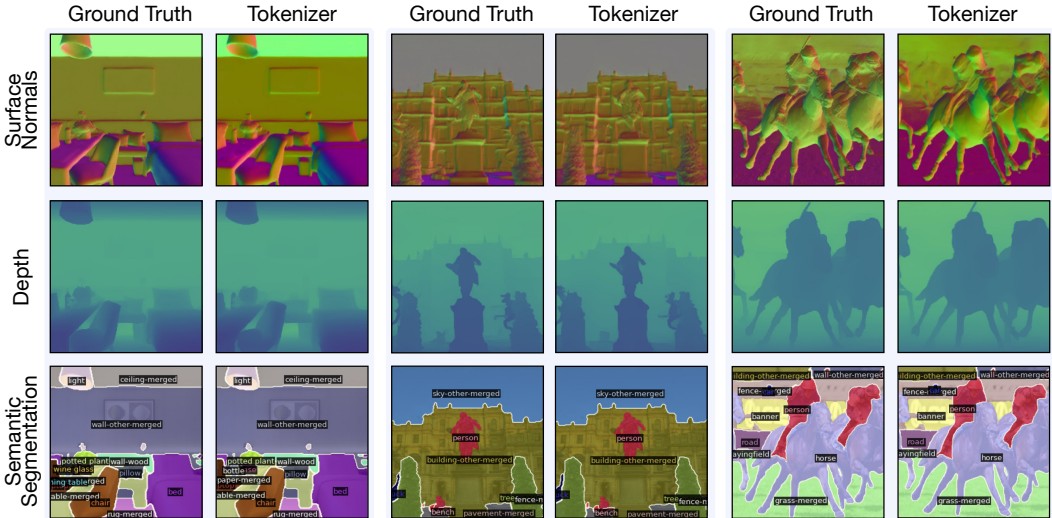

Figure 20: **Tokenizer reconstructions:** We show sample reconstructions of surface normals, depth, and semantic segmentation pseudo labels from the CC12M validation set at $224 \times 224$ resolution ($14 \times 14$ tokens). Quantitative evaluations are provided in Table 31 (last row), confirming that the reconstruction quality does not bottleneck 4M's out of the box performance.

## F.2    Text-to-image performance

In this section, we evaluate 4M's text-to-image capabilities, with detailed quantitative results shown in Table 32. To perform a controlled comparison, we train a pure text-to-image variant of 4M-B, akin to Muse [18], for a total of 300B tokens on CC12M, and using the same RGB tokenizer as the 4M models. 4M trained on all modalities achieves comparable FID and CLIP scores to this specialist model, and at the same time can be conditioned on any pre-training modality and can solve several common vision tasks out of the box. We also compare against the $512 \times 512$ base model of Stable Diffusion 2.1 (SD-2.1) [95] and observe a considerable gap to SOTA generative models on OOD data. It is important to highlight that SD-2.1 was trained on datasets that were two orders of magnitude larger and with a compute budget one order of magnitude greater than 4M-XL. When taking into account the scaling behaviors of comparable token-based text-to-image models such as Muse or MAGE, we anticipate a substantial improvement in the generation quality of 4M models if they were trained under a similar data and computational regime, alongside the use of improved image tokenizers.

Table 32: **Quantitative evaluation of text-to-image generative capabilities:** We evaluate the text-to-image capabilities of 4M and compare them with a controlled text-to-image version of 4M (similar to Muse), and Stable Diffusion 2.1-base. We compute FID and CLIP-L/14 scores on 30K images of the CC12M and COCO validation sets after resizing the generations to 256x256. All runs use guidance scale 3.0. In-domain, 4M can approach SD-2.1 and matches the Muse baseline in a controlled setting. On COCO, we observe a larger gap to SD-2.1 which was trained on orders of magnitude more data and compute.

| Model | Res. | Dataset | A100-hours | Text encoder | CC12M val (30K) | | COCO val (30K) | |
|---|---|---|---|---|---|---|---|---|
| | | | | | FID ↓ | CLIP score ↑ | FID ↓ | CLIP score ↑ |
| 4M-B | 224 | CC12M | 2.3k | - | 15.9 | 20.6 | 37.5 | 21.4 |
| 4M-L | 224 | CC12M | 9.2k | - | 11.9 | 18.9 | 30.1 | 23.1 |
| 4M-XL | 224 | CC12M | 24.5k | - | 10.7 | 22.1 | 27.0 | 23.7 |
| 4M-B "Muse" | 224 | CC12M | 1.6k | - | 18.3 | 18.8 | 39.8 | 20.3 |
| SD-2.1-base | 512 | Curated LAION-5B | > 200k | CLIP ViT-L/14 | 9.1 | 25.5 | 10.1 | 25.5 |

# G  Broader Impact

## G.1  Computational costs

The training time of the `4M` models used in the transfer and generative results depends on their size: `4M-B` was trained in 1.5 days on 64 A100 GPUs, `4M-L` was trained in 3 days on 128 A100 GPUs, and training `4M-XL` took 8 days on 128 A100 GPUs.

The reference model used in the ablations can be trained in 12 hours on 32 A100 GPUs, or 2 days on 8 A100 GPUs. Transferring a pre-trained model to the 35 benchmark tasks used for the ablations takes approximately 3 days on 8 A100 GPUs.

## G.2  Social impact

We developed a framework for training general-purpose foundation models that, as demonstrated, can be conveniently re-purposed for various tasks a practitioner may be interested in. We also committed to open-sourcing our code and models. These actions support the public with the democratization of the tools and the possibility of transparent inspection and safeguarding. While our model is not particularly poised for negative use compared to the alternatives, it should be noted that powerful generative models are a general tool and have the potential to be used in ways authors did not intent. In addition, the data they are trained on may incorporate various societal biases or contain samples gathered in different ways from the internet. We trained our models on CC12M [19] which is an open-sourced dataset and has been curated to some degree (e.g., people's names are redacted), yet, due to the imperfections in this process, we still advise caution when using the models for generative purposes.

