# OpenReview forum: "4M: Massively Multimodal Masked Modeling"
_NeurIPS.cc/2023/Conference — NeurIPS 2023 spotlight_

### Official Review · Reviewer_q9W9 · 2023-06-28

**Soundness:** 3 good
**Presentation:** 3 good
**Contribution:** 4 excellent
**Rating:** 7
**Confidence:** 4

**Summary:**

The paper addresses the importance of a versatile model that is not limited to single modality and task and proposes a multi-modal pre-training scheme called 4M. 4M is a single encoder-decoder architecture trained on a large set of image and sequence-like modalities. The modalities including text, images, geometric and semantic are brought into a joint representation as tokens through modality-specific tokenizers. The training procedure relies on multi-modal masked training, with only a small set of tokens used as inputs and targets. Extensive experimentations showed that 4M can solve many common vision tasks out of the box, can be fine-tuned to unseen tasks as well as perform multi-modal controllable generation.

**Strengths:**

This publication has several strengths including:

1) The writing is very clear and easy to understand.

2) The proposed approach is scalable across three key aspects - data (more training samples increase performance), architecture (improve performance with model size as well as remain stable) and training objective (handle growing number of modalities without incurring excessive computational costs).

3) Good experimental methodology with carefully designed ablations that justifies architectural design decisions especially impacts of  input modalities and target tasks, multi-modal masking strategy as model and data scaling.

4) Very exhaustive in-depth experimentation showcasing the key capabilities - zero-shot generalization to diverse set of vision tasks, fine-tune to unseen tasks, multi-modal controllable generation.

**Weaknesses:**

1) The paper seemingly lacks any quantitative evaluation of its generation capabilities or comparison with existing state-of-the-art methods. I'd appreciate it if the authors can elaborate on this.

2) Another concern is that the paper lacks any discussion on the robustness of the proposed approach to the quality of the datasets, since low-quality data is usually readily available compared to high-quality data. This will be critical for data scaling as well as will align with model scaling to 4M-XL and beyond.

3) Minor comment: I am curious if the authors have performed any out-of-distribution analysis.

**Questions:**

The paper in its current form lacks discussion on the robustness of 4M to dataset quality as well as evaluation of its generation capabilities. Please refer to the “Weakness” section for details.

**Limitations:**

Yes, the authors discuss the limitations in the paper.

---

> ### Author Rebuttal · Authors · 2023-08-10
>
> We thank reviewer q9W9 for their positive feedback. We address the main concerns and questions in following response:
>
> > The paper seemingly lacks any quantitative evaluation of its generation capabilities or comparison with existing state-of-the-art methods. I'd appreciate it if the authors can elaborate on this.
>
> For a quantitative analysis of 4M’s generative capabilities, please see the section 2 of the common response and the `PDF`.
>
> > Another concern is that the paper lacks any discussion on the robustness of the proposed approach to the quality of the datasets, since low-quality data is usually readily available compared to high-quality data. This will be critical for data scaling as well as will align with model scaling to 4M-XL and beyond.
>
> Assessing the influence that the “quality” of a dataset has on a pre-training strategy is a **highly interesting and important question**. Works like DataComp [1] aim to achieve this by developing filtering strategies for multi-modal datasets, but it is an **open question and research direction of its own**.
> To ablate the influence of the choice of pre-training dataset to some extent, **we pseudo labeled ImageNet-21K, as well as a 15M subset of COYO-700M** [2], and trained 4M-B models on each. Tab. 3 in the rebuttal `PDF` shows that the 4M-B models trained on these datasets achieve a **similar performance to the CC12M version, while still surpassing previously reported baselines**.
> To add to that, and related to reviewer RJ3x’s question on pseudo labeling quality, we argue that **the use of pseudo labeling is a strength of the approach as it is inherently more scalable than using high-quality off-the-shelf datasets**.
> Similar to the common practice in NLP of pre-training on large uncurated datasets and tuning the model with higher-quality data, it is conceivable that similar approaches can work well for 4M too.
>
> > Minor comment: I am curious if the authors have performed any out-of-distribution analysis.
>
> Please see Tab. 4 in the rebuttal `PDF` for an OOD analysis of several ImageNet-1k transfers to IN-A, IN-R, IN-S, IN-C, and IN-3DCC. **4M shows strong robustness to various OOD domains and corruptions, and is competitive with DeiT III**, which is a specialist model.
> Further, we show the **strong zero-shot performance of 4M** on surface normal, depth, and semantic segmentation in Tab. 1 in the rebuttal `PDF`. The performance is evaluated on the DIODE and COCO datasets which are not part of the 4M training dataset. **We note that on this OOD data, 4M matches or even surpasses the pseudo labeler networks and other strong baselines.**
>
> [1] DataComp: In search of the next generation of multimodal datasets, Gadre et al., 2023
>
> [2] https://github.com/kakaobrain/coyo-dataset

---

> > ### Comment · Reviewer_q9W9 · 2023-08-20
> >
> > I appreciate the authors addressing my raised concerns about the quantitative evaluation of 4M's generation capabilities and also providing insights on the effect of dataset quality on pre-training strategy. I suggest the authors add the above results to the revised paper. I am happy to increase my rating.

---

### Official Review · Reviewer_QEFU · 2023-07-04

**Soundness:** 2 fair
**Presentation:** 2 fair
**Contribution:** 1 poor
**Rating:** 5
**Confidence:** 5

**Summary:**

The paper proposes a multi-modal masked modeling pre-training scheme (4M) that unifies a several modalities – including text, images, geometric, and semantic modalities, neural network feature maps. The tokenization and masked modeling enable the efficient pretraining of 4M. The pretrained model can 1) achieve reasonable finetuned performance on vision tasks, 2) achieve conditional generation under different modalities.

**Strengths:**

This work is a good practice on multi-modal masked modeling pre-training and achieve reasonable performance on both finetuned downstream tasks and generative tasks.

**Weaknesses:**

1.	This work is a combinational work of existing methods and lacks technical novelty.
-	The multi-modal masked modeling has been utilized by various existing works. E.g. MultiMAE has proven the feasibility of multi-modal masked modeling. Vision-NLP multi-modal is also explored by exsting works,e.g. MAGVLT.
-	The tokenization is claimed to be a key part for the efficient pretraining of 4M, while the tokenization has been widely used existing works, and the tokenizers used in this work are mostly borrow from other works.
-	The major contribution seems to be the the multi-modal pretrained data based on the CC12M dataset. But this is no novelty on the pseudo-labeling of the data.
2.	The performance is not impressive given the large-scale data used in this work.
-	The paper claims 4M can perform a diverse set of vision tasks out of the box, but no experimental results are showed in the manuscript.
-	In Tab.1, the finetuned performance is relatively weak compared other pretrained methods. In addition, many stronger pretrained methods are not included in this paper.
-	The conditional generation results are interesting. While there is no quantitative performance comparison with other methods such as contorlnet. Also, one drawback of 4M based conditional generation is that the model accepts fixed modal after pertraining, while controlnet can be easily extended to new modal.
-	The paper claims the pretraining efficiency is an advantage, yet no experimental result is given to prove this point.

**Questions:**

1.	Clarification of the novelty of this paper.
2.	The performance comparison and experimental results to support the claims in this paper.

**Limitations:**

The limitation of this work is mainly on the lacking novelty and the experimental results cannot well support the claims.

---

> ### Author Rebuttal · Authors · 2023-08-10
>
> We thank reviewer QEFU for their feedback. We address the main concerns and questions in following response:
>
> > This work is a combinational work of existing methods and lacks technical novelty.
> The multi-modal masked modeling has been utilized by various existing works. E.g. MultiMAE has proven the feasibility of multi-modal masked modeling. Vision-NLP multi-modal is also explored by exsting works,e.g. MAGVLT.
> The tokenization is claimed to be a key part for the efficient pretraining of 4M, while the tokenization has been widely used existing works, and the tokenizers used in this work are mostly borrow from other works.
> >
> First, we want to clarify that the simplicity of our method is a highly desirable property given its competitiveness and novel capabilities. Nevertheless, **our approach includes several technical innovations**, which we believe distinguish 4M from existing methods in meaningful ways.
> - **Masking:** While 4M's multi-modal masking strategy is inspired by MultiMAE, we introduce several key changes that are crucial for scaling our models beyond the three image-like modalities of MultiMAE. For a comprehensive overview of these changes, please see our response to reviewer f31a. Additionally, please note that MAGVLT, although relevant, focuses only on text-image pairs and was published within two months of the submission deadline, making it concurrent work.
> - **Tokenization:** The novelty lies in 4M's ability to work with multiple modality-specific tokenizers. Unlike methods like Unified-IO, which operates on a single RGB image tokenizer, our approach enables scaling to modalities beyond those that can be represented as images, including neural network feature maps. This key distinction is not about the tokenizers themselves but how 4M leverages them to jointly operate on diverse modalities.
> - **Architecture:** 4M's architecture was intentionally designed to be as close as possible to a standard Transformer encoder-decoder to take advantage of their scalability and flexibility. However, we also had to include some crucial modifications to enable joint modeling of both image-like and sequence-like modalities within a single encoder-decoder architecture, as described in Section 2.2 of our paper.
> - **Importance of combinational work:** It's worth noting that the act of bringing together these specific methods in itself introduces novelty. **Our combination of techniques leads to new capabilities and improved results that wouldn't be possible otherwise**. For example, unlike MultiMAE which can’t be used for generative tasks due to lack of tokenization, 4M can function as a generative model while also showing much better transfer performance.
>
> > The paper claims 4M can perform a diverse set of vision tasks out of the box, but no experimental results are showed in the manuscript
> >
> Please see Table 2 of the rebuttal `PDF` where we show the out of the box (zero-shot) performance of 4M on surface normals, depth, and semantic segmentation on the DIODE and COCO datasets. On this data, **4M matches or even surpasses the pseudo labeler networks and other strong baselines**.
>
> > In Tab.1, the finetuned performance is relatively weak compared other pretrained methods. In addition, many stronger pretrained methods are not included in this paper.
> >
> In our transfer study (Tab. 1 of our paper), **the pretraining methods used as comparison are recognized as strong models**. For example, MAE serves as the backbone for task-specific foundation models such as ViTDet [1] and SAM [2].  Note that 4M outperforms all reported baselines including MAE on all tasks except ImageNet classification.
>
> Furthermore, in Tab. 3 of the rebuttal `PDF`, we provide a comparison with DINOv2-Base, one of the strongest publicly available ViT-B models. However, please note that these models are not directly comparable as DINOv2:
>
> 1. uses an **order of magnitude more training data.** Furthermore, the data was curated to be similar to evaluation datasets, including those that used in our downstream tasks (IN1K, ADE20k, NYUv2). Thus, there is less of a distribution shift for DINOv2 when transferring with these datasets.
>
> 2. requires **orders of magnitude more compute.**
>
> 3. is distilled from a significantly larger model (DINOv2 ViT-g, 1.1B params), which gives a boost in performance compared to training from scratch.
>
> While DINOv2 is able to attain slightly better performance on downstream tasks, 4M-B is still able to approach DINOv2’s performance on several tasks despite notable differences in computational cost and dataset size.
> | Model | Dataset size | Compute cost (A100 hrs) |
> | --- | --- | --- |
> | 4M Base | 12M | 2300 |
> | DINOv2 Base Distilled (from ViT-g) | 142M | 22000 + 5300 |
>
> > The paper claims the pretraining efficiency is an advantage, yet no experimental result is given to prove this point.
> >
> The 4M training scheme produces a model that can predict any task from any subset of full or partial modalities — all in a single and highly efficient pre-training run. Input and target masking are crucial to make this work efficiently. Already MAE has shown that dropping masked tokens at the input level can significantly improve pre-training efficiency. MultiMAE further demonstrated its importance for multi-modal training, but as the number of tasks increases, so does the computational cost of the decoders since all masked and non-masked patches of all modalities are always decoded. 4M addresses this issue by decoding only a random subset of the masked tokens (as ablated in Appendix Tab. 17), at no cost to downstream performance.
>
> Furthermore, in Fig. 2 of the rebuttal `PDF`, **we quantitatively demonstrate that target masking can significantly improve training efficiency**, especially when training on a large number of modalities or on modalities with a large sequence length.
>
> [1] Exploring Plain Vision Transformer Backbones for Object Detection, Li et al., 2022
>
> [2] Segment Anything, Kirillov et al., 2023

---

> > ### Comment · Reviewer_QEFU · 2023-08-19
> > **Thanks for your response.**
> >
> > - After seeing the response, my major concern about the novelty remains.
> > - MAE cannot be regarded as a strong baseline now since many pretraining methods have been proposed with much stronger performance.
> > - On the good side, this work is a good practice to combine exsiting methods to achieve the multi-modal pretraining.
> > - Considering the multi-modal pretraining is a promising, I would like to rise my rating.

---

> > > ### Author Response · Authors · 2023-08-21
> > > **Rating update**
> > >
> > > We are glad to hear that the reviewer QEFU found the proposed multi-modal training promising and would like to increase their rating. We kindly remind the reviewer that the deadline is soon and the rating update needs to be done via editing the original review. We thank once again for their feedback which improved the quality of our work.

---

### Official Review · Reviewer_qLD7 · 2023-07-06

**Soundness:** 4 excellent
**Presentation:** 4 excellent
**Contribution:** 4 excellent
**Rating:** 9
**Confidence:** 4

**Summary:**

The paper presents a foundation model for a variety of vision tasks. The authors show it can perform many key vision tasks out of the box and can also be fine-tuned to achieve highly competitive performance on unseen downstream tasks and input modalities. To handle the variety of modalities, the inputs/outputs are encoded into sequences of discrete tokens, and the model is trained on all the tasks simultaneously via a multi-modal masked modeling objective.

**Strengths:**

The paper presents strong results and a scalable method to perform a variety of vision tasks. The ablation study covers almost all the aspects of the model. The results indeed prove the superiority of multimodal training over the baselines, without any need for augmentations. The paper is well-written and paves the way for a variety of research questions about the interactions between different modalities.

**Weaknesses:**

While to tokenization method allows the model to train on a variety of tasks with a single architecture and cross-entropy loss, it also introduces quantization of the space of inputs/outputs. While this quantization does not harm the results for text, it might decrease the quality of the results for other domains (the segmentation boundaries might not be fully aligned with the objects for example). This idea induces an upper bound on the performance of such an algorithm and should be discussed. One way to evaluate this upper bound is by encoding and decoding back the ground-truth results of different domains (e.g. - quantizing the ground-truth segmentation masks and decoding them back), to verify the reconstruction quality and the downstream task-specific performance.



**Questions:**

One suggestion is to mention in the related work other papers that deal differently with multimodal inputs/outputs for solving various vision tasks altogether. This line of work includes:

- Wang et al., "Images Speak in Images: A Generalist Painter for In-Context Visual Learning", CVPR'23

- Bar et al. "Visual prompting via image inpainting", NeurIPS'22


One question that I had (and I am not sure how to evaluate) - what is more helpful for downstream performance - inputting during training tokens that correspond to the same image position but from different domains, or using more tokens from the same domain but from different locations in the image?

**Limitations:**

The limitations of the paper are discussed and addressed in the last section.

---

> ### Author Rebuttal · Authors · 2023-08-10
>
> We thank reviewer qLD7 for their positive feedback. We address the main concerns and questions in following response:
>
> > While this quantization does not harm the results for text, it might decrease the quality of the results for other domains (the segmentation boundaries might not be fully aligned with the objects for example). This idea induces an upper bound on the performance of such an algorithm and should be discussed. One way to evaluate this upper bound is by encoding and decoding back the ground-truth results of different domains (e.g. - quantizing the ground-truth segmentation masks and decoding them back), to verify the reconstruction quality and the downstream task-specific performance.
>
> Thank you for the suggestion. We **measure the reconstruction quality of the surface normal, depth, and semantic segmentation tokenizers** on 5000 CC12M validation images at a resolution of 224x224. Note that we cannot evaluate the performance of the tokenizers on datasets like DIODE or OASIS due to the presence of masks/holes in the dense labels. Since the tokenizers were trained on pseudo labeled data, they do not handle such masks. This is, however, not an inherent limitation, as the tokenizers could be trained with masked inputs, which we did for the Taskonomy and Hypersim tokenizers in the ablations.
> Fig. 1 in the rebuttal `PDF` shows qualitative examples and Tab. 1 (last row) shows reconstruction metrics. While measuring the reconstruction on CC12M is not fully comparable to the DIODE and COCO zero-shot performance of 4M and baselines shown in Tab. 1, we note that **the tokenizer reconstruction errors are of a significantly lower magnitude than the prediction errors from RGB**. This indicates that the tokenization is not a strong bottleneck when considering the difficulty of predicting these tasks from RGB. That said, tokenization may remove or change fine details present in images, which may not be as clear in these metrics but can be visually apparent (see Fig. 1). A remedy to this could be to perform tokenization on higher resolution images or to decrease the patch size. We also generally expect future advances in tokenizer training to translate directly to zero-shot and downstream performance improvements.
>
> > […] mention in the related work other papers that deal differently with multimodal inputs/outputs for solving various vision tasks altogether.
>
> Thank you for the suggestions, we’ll make sure to include a discussion of works that unify different vision tasks in this way in the camera ready version upon acceptance.
>
> > […] what is more helpful for downstream performance - inputting during training tokens that correspond to the same image position but from different domains, or using more tokens from the same domain but from different locations in the image?
>
> This is indeed an interesting question, and our ablation of the input and target mask sampling parameters $\alpha$ (see Appendix Tab. 15) provides a partial answer. Setting the input and target alphas to a very low value (e.g. 0.1) corresponds to sampling the number of tokens per modality from a “spiky” Dirichlet distribution, i.e. most of the time tokens are sampled from single modalities. This is close to the latter case mentioned, i.e. using more tokens from the same domain but from different locations in the image and predicting across domains. **This setting performs slightly worse on downstream transfers compared to more random mask sampling approaches (alphas >= 0.2), or mixtures of sampling strategies (see Appendix Tab. 18).**
> Note that on the other hand, the use of higher alphas (> 1.0) does not correspond to the former case (sampling tokens that correspond to the same image position but from different domains). Restricting the sampling in that way has been studied in masked video pre-training, notably spatio-temporal MAEs [1] and VideoMAE [2], and it has been observed that more random masking strategies perform similar or better. Extending this analysis to the multi-modal case is interesting future work.
>
> [1] Masked Autoencoders As Spatiotemporal Learners, Feichtenhofer et al., 2022
>
> [2] VideoMAE: Masked Autoencoders are Data-Efficient Learners for Self-Supervised Video Pre-Training, Tong et al., 2022

---

> > ### Comment · Reviewer_qLD7 · 2023-08-21
> >
> > Thanks for the response. I don't have further comments. I will keep my rating.

---

### Official Review · Reviewer_f31a · 2023-07-06

**Soundness:** 3 good
**Presentation:** 4 excellent
**Contribution:** 4 excellent
**Rating:** 7
**Confidence:** 5

**Summary:**

The paper presents a unified transformer model by using an effective multi-modal pre-training scheme. The authors propose to perform masked modeling across different modalities. This is made possible by unifying the representation space of the considered modalities by mapping them into discrete tokens and then performing multi-modal masked modeling on a small subset of tokens. Experimental results demonstrate several promising results.

**Strengths:**

- This paper is technically valid and interesting. By conditioning on arbitrary modalities, the model can have great potential for a variety of multimodal intelligence capabilities.
- The authors present comprehensive experiments and ablations, providing insightful discussions. The paper can be a good reference for future researchers.
- The paper is well-written and easy to follow.

**Weaknesses:**

- The multi-modal masking strategy is highly similar to prior works, like MultiVAE. The mask-modeling part of this paper is somewhat less interesting and less innovative. The innovation is more in the developed system framework.
- I don't find other significant concerns in the proposed method.

**Questions:**

N/A

**Limitations:**

The authors discussed some of the limitations, but this is more like descriptions of future work. No method is developed in address the limitations.

---

> ### Author Rebuttal · Authors · 2023-08-10
>
> We thank reviewer f31a for their positive feedback. We address the main concerns and questions in following response:
>
> > The multi-modal masking strategy is highly similar to prior works, like MultiVAE. The mask-modeling part of this paper is somewhat less interesting and less innovative. The innovation is more in the developed system framework.
>
> While 4M’s multi-modal masking strategy is inspired by MultiMAE, 4M proposes several important changes that enable further scaling of multi-modal models beyond three image-like modalities:
>
> - MultiMAE’s masking strategy assumes that all modalities are image-like, in their case RGB, depth and semantic segmentation maps. Applying the same masking strategy to sequence modalities such as captions or bounding boxes would not allow 4M to generate these modalities at inference time. We therefore propose to **use span-masking [1] to both benefit from masked pre-training and enable generation of sequence modalities**.
> - MultiMAE requires separate decoders for each target modality, and **always decodes all input and all mask tokens**. The overhead of this is manageable with three modalities and shallow decoders, but may not scale to much larger number of modalities. We propose target masking as a means to overcome this. For example, if we trained 4M without target masking, the number of tokens to decode would be around 1000, which would incur a significant compute overhead (see rebuttal `PDF` Fig. 2). In addition, our ablations in Appendix Tab. 17 show that **lower target masking budgets are more compute efficient** (for a fixed number of total training tokens).
> - In addition, we propose **mixtures of masking strategies** to train 4M on a diverse set of inputs and targets (see Appendix Tab. 18), with the resulting models striking a compromise between either masking scheme.
>
> > The authors discussed some of the limitations, but this is more like descriptions of future work. No method is developed in address the limitations.
>
> In the Conclusion and Limitations section of the main paper **we discuss several limitations** (limited number of modalities, tokenizer quality, dataset size and quality) and **propose potential solutions** to address them. In addition, we list here several other limitations and possible ways to overcome them:
>
> - **Text understanding**: Recent state of the art text-to-image models like Imagen [2], Parti [3], Muse [4], or Stable Diffusion 2.1 [5] commonly train on powerful text encoders (e.g. T5-XXL or OpenCLIP-ViT/H) instead of classical text tokenizers to significantly improve their image generation fidelity and text understanding. 4M is trained directly on the text tokenizer, but we expect similar text-to-image improvements when training on LLM embeddings instead.
> - **Fine-grained editing**: Since 4M operates on discrete sets of tokens, in-painting is roughly constrained to the relatively coarse grid of tokens. Since every token affects an area slightly larger than its relative size on the image, the area that needs to be selected to remove a certain object is even larger. In addition, tokenization is a lossy process and destroys fine-grained details. Unlike diffusion models, we are not able to perform pixel-level edits. To remedy this, we can consider in-painting at a higher resolution (with a finer grid of tokens) or training specialized in-painting adapters.
> - **Alignment to downstream objectives**: 4M is trained by simply minimizing the cross-entropy loss between predicted and ground truth tokens. If this objective is misaligned with a certain downstream objective (e.g. aesthetics or a downstream task metric), we are not optimizing what we care for directly. To address this, we can consider fine-tuning the pre-trained 4M model using reinforcement learning on downstream objectives that are otherwise hard to optimize for directly. [6]
> - **Flexible image resolution**: As most other models, 4M is trained on square images of a fixed resolution — in our case 224x224. We trained a token super-resolution model that can map 4M outputs to 448x448, but to work out of the box at different resolutions and aspect ratios, we can consider approaches similar to FlexiViT [7] or NaViT [8]. This can be done as a fine-tuning step after training the model at the base resolution.
>
> [1] Exploring the Limits of Transfer Learning with a Unified Text-to-Text Transformer, Raffel et al., 2019
>
> [2] Photorealistic Text-to-Image Diffusion Models with Deep Language Understanding, Saharia et al., 2022
>
> [3] Scaling Autoregressive Models for Content-Rich Text-to-Image Generation, Yu et al., 2022
>
> [4] Muse: Text-To-Image Generation via Masked Generative Transformers, Chang et al., 2023
>
> [5] https://huggingface.co/stabilityai/stable-diffusion-2-1-base
>
> [6] Tuning computer vision models with task rewards, Pinto et al., 2023
>
> [7] FlexiViT: One Model for All Patch Sizes, Beyer et al., 2022
>
> [8] Patch n' Pack: NaViT, a Vision Transformer for any Aspect Ratio and Resolution, Dehghani et al., 2023

---

> ### Comment · Reviewer_f31a · 2023-08-20
>
> Thanks for the response. I don't have further comments. I will keep my rating.

---

### Official Review · Reviewer_RJ3x · 2023-07-13

**Soundness:** 3 good
**Presentation:** 3 good
**Contribution:** 3 good
**Rating:** 6
**Confidence:** 5

**Summary:**

This paper proposes a multimodal pre-training framework named 4M, which employs the masked data modeling style to train a transformer encoder-decoder archtecture that is capable of performing different downstream tasks. Experiments show that 4M delivers competitive transfer ability on these tasks compared with MAE / DEiT III / BEiT v2.

**Strengths:**

1. The baseline settings in the experiments are fair and sound, especially the self-baselines to control other variables.
2. The ablation studies provide insightful discussions on the design choices of the pre-training strategy.

**Weaknesses:**

1. The multi-modal and multi-task training of 4M needs datasets with all required modalities and labels. However, this kind of well-annotated dataset is hard to obtain and not scalable. This research employs pseudo labeling to extend existing image-text datasets such as CC12M. Therefore, the performance of the off-the-shelf labelers are important. The authors should provide more detailed and careful discussions and ablations on that.
2. The downstream tasks are limited, especially considering the target of this paper, i.e. ``massively pre-training''. Quantative results on more diverse tasks / datasets should be examined, especially on the transfer ability to novel tasks.

**Questions:**

1. In the Table.2 of ablation studies, why use loss instead of corresponding task metrics as the mesure?
2. Could the authors provide some quantative results on the generative capabilities of 4M?

**Limitations:**

The model size and the data size could be further scaled up. The authors have discussed some limitations in their paper.

---

> ### Author Rebuttal · Authors · 2023-08-10
>
> We thank reviewer RJ3x for their positive feedback. We address the main weaknesses, questions, and limitations in the following:
>
> > The multi-modal and multi-task training of 4M needs datasets with all required modalities and labels. However, this kind of well-annotated dataset is hard to obtain and not scalable. This research employs pseudo labeling to extend existing image-text datasets such as CC12M. Therefore, the performance of the off-the-shelf labelers are important. The authors should provide more detailed and careful discussions and ablations on that.
>
> - We generally agree with this summary. However, **Pseudo labeling is inherently more scalable than other approaches** due to the high availability of RGB images and off-the-shelf models. Combining datasets with incomplete annotations (like UnifiedIO [1] does) is limited by the relatively smaller number of those datasets, and the end result is not an aligned large dataset -- however, our method is not incompatible with that approach (even for the fine-tuning phase), and it'd be an interesting experiment to run. Existing annotated multi-task datasets like Taskonomy [2], Omnidata [3], Hypersim [4], etc, are also too limited in terms of their domain. Overall, Pseudo labeling is a good enabling strategy at the moment, and we acknowledge the value of training on incomplete data for the future. We will update the camera-ready with a discussion.
> - Performance of the off-the-shelf labelers: As shown in Tab. 1 in the rebuttal `PDF`, **the larger the 4M model, the more it approaches the performance of the original pseudo labeler, and may even slightly surpass it**. Indeed, we would expect higher-quality pseudo labels to translate well to better zero-shot performance. In addition, we use pseudo labeling to enable massively multi-modal pre-training, but it is conceivable that a fine-tuning stage with a much smaller dataset of high-quality data can significantly improve zero-shot performance. This practice of pre-training on large-scale, noisy and uncurated data and then tuning the model on a clean dataset or using reinforcement learning has worked well in the field of natural language processing (e.g., ChatGPT, or GPT-4), and we expect a similar approach to work well for multi-modal foundation models too.
>
> > The downstream tasks are limited, especially considering the target of this paper, i.e. ``massively pre-training''. Quantitative results on more diverse tasks / datasets should be examined, especially on the transfer ability to novel tasks.
>
> We agree that the transfer learning study could benefit from a more diverse set of novel downstream tasks. That said, our ablation that contains **data- and compute-controlled baselines of MAE (RGB->RGB) and BEiT v2 (RGB->CLIP) style models** contains an extensive set of both **novel downstream tasks and datasets, spanning 35 tasks over 4 different datasets**. Many of the transfers contain input modalities and target tasks that were **not seen during 4M pre-training**. 4M pre-trained on all modalities performs **better than the baselines** on most of these tasks.
>
> > In Table.2 of ablation studies, why use loss instead of corresponding task metrics as the measure?
>
> The aim of Tab. 2 is to show **how well certain instantiations of 4M transfer to arbitrary new distributions of tokens**. Since downstream performance comes down to A) how well the tokens are able to represent the downstream tasks (i.e. tokenizer reconstruction error), and B) how well 4M is able to predict these tokens, we abstract away A for this ablation since the tokenizers are the same for all settings, and only report B. In addition, reporting the cross-entropy losses, or equivalently log-perplexity, is a standard practice in NLP, and aligns well with the aim of this ablation and with the way 4M is pre-trained and fine-tuned. Reporting the cross-entropy loss makes **comparison more uniform** and avoids having to scale and average wildly different task-specific metrics such as mAP, mIoU, MSE, etc. that are not comparable with each other.
>
> > Could the authors provide some quantitative results on the generative capabilities of 4M?
>
> For a quantitative analysis of 4M’s generative capabilities, please see the section 2 of the common response and the `PDF`.
>
> > The model size and the data size could be further scaled up. The authors have discussed some limitations in their paper.
>
> Scaling the model and dataset size beyond 4M-XL is out of the scope for this paper, but we agree that 4M could benefit from exploration in that area. Our scaling trends (Fig. 5 in the main paper) and evidence from token-based text-to-image models like Muse [5] and Parti [6] paint a promising picture and suggest scaling the data and model size to be an exciting future direction.
>
> [1] Unified-IO: A Unified Model for Vision, Language, and Multi-Modal Tasks, Lu et al., 2022
>
> [2] Taskonomy: Disentangling Task Transfer Learning, Zamir et al., 2022
>
> [3] Omnidata: A Scalable Pipeline for Making Multi-Task Mid-Level Vision Datasets from 3D Scans, Eftekhar et al., 2021
>
> [4] Hypersim: A Photorealistic Synthetic Dataset for Holistic Indoor Scene Understanding, Roberts et al., 2020
>
> [5] Muse: Text-To-Image Generation via Masked Generative Transformers, Chang et al., 2023
>
> [6] Scaling Autoregressive Models for Content-Rich Text-to-Image Generation, Yu et al., 2022

---

> > ### Comment · Reviewer_RJ3x · 2023-08-21
> > **Thanks for the rebuttal**
> >
> > Thank the authors for their response. I maintain my score as "weak accept".

---

### Author Rebuttal · Authors · 2023-08-10

# Response to all reviewers

We thank the reviewers for their insightful comments and are glad to hear that the reviewers found the paper to be **“technically valid and interesting”** with **“great potential for a variety of multimodal intelligence capabilities”** (f31a), provide **“insightful discussions on the design choices of the pre-training strategy”** (RJ3x), and appreciate the **“very exhaustive in-depth experimentation showcasing the key capabilities”** (q9W9). We are also glad that the reviewers recognized the **“strong results and a scalable method to perform a variety of vision tasks”** (qLD7) and commended our writing as being **“very clear and easy to understand / follow”** (q9W9, f31a) / **"well-written and paves the way for a variety of research questions about the interactions between different modalities"** (qLD7).



## 1. Additional results overview
We address the reviewers’ remaining questions and concerns in the individual responses and rebuttal `PDF`. We discuss general questions on the generative capabilities below the following list of new experiments and major addressed questions:
- QEFU: Out of the box (zero-shot) performance evaluation (`PDF` Tab. 1)
- qLD7: Tokenizer reconstruction quality (`PDF` Tab. 1 and Fig. 1)
- RJ3x, QEFU, q9W9: Quantitative evaluation of generative capabilities (`PDF` Tab. 2)
- QEFU, q9W9: Comparison to strong baselines and robustness to dataset quality (`PDF` Tab. 3)
- q9W9: OOD analysis (`PDF` Tab. 4)
- QEFU: Pre-training efficiency (`PDF` Fig. 2)

## 2. Common questions

> RJ3x, QEFU, q9W9: General quantitative evaluation of generative capabilities

Please see Tab. 2 in the rebuttal `PDF` for a quantitative comparison of 4M across model sizes, a controlled text-to-image baseline, as well as Stable Diffusion 2.1. The metrics shown are computed on 30k subsets of CC12M and COCO validation sets, and we interpolate all generated images to 256x256.
To perform a controlled comparison, we train a pure text-to-image variant of 4M-B, in spirit similar to Muse [1], for a total of 300B tokens on CC12M, and using the same RGB tokenizer as used for 4M. **4M trained on all modalities achieves comparable FID and CLIP scores to this specialist model, and at the same time can be conditioned on any pre-training modality and can solve several common vision tasks out of the box.**
We also compare against the 512x512 base model of Stable Diffusion 2.1 (SD-2.1) [2] and observe a considerable gap to SotA generative models on OOD data. We note here, however, that **SD-2.1 was trained on datasets two orders of magnitude and a compute budget one order of magnitude larger than what was used to train 4M-XL**. 4M is a general multi-modal pre-training strategy and considering the scaling curves of similar token-based text-to-image models like Muse or MAGE, we expect 4M’s generation quality to significantly improve given a similar data and compute regime, and better image tokenizers.

> QEFU: Also, one drawback of 4M based conditional generation is that the model accepts fixed modal after pertraining, while controlnet can be easily extended to new modal.

Adapting multi-modal Transformers like 4M to new modalities / tasks is an exciting future research direction. Parameter-efficient fine-tuning techniques like Low-Rank Adaptation (LoRA) have been shown to work well on LLMs and diffusion models, and we would expect similar techniques to allow 4M to be efficiently adapted to additional modalities. Since our pre-training dataset is pseudo labeled, this would require only a dataset of the new modality and RGB images, which would also be the case for training a new ControlNet.

[1] Muse: Text-To-Image Generation via Masked Generative Transformers, Chang et al., 2023

[2] https://huggingface.co/stabilityai/stable-diffusion-2-1-base

---

### Decision · Program_Chairs · 2023-09-21

**Decision:**

Accept (spotlight)

**Comment:**

This paper presents 4M: A single unified Transformer encoder-decoder trained using masked modeling objective across a wide range of modalities including text, images, geometric, and semantic modalities. The paper received nearly unanimous accept reviews, including 1x Weak accept, 2x accept, and 1x very strong accept. The reviewers appreciate the well written paper with extensive analysis and experimentation, insightful discussions, and an interesting and scalable approach. ACs concur with the reviewer consensus.